# Are Defenses for Graph Neural Networks Robust?

**Felix Mujkanovic**[1*], **Simon Geisler**[1*], **Stephan Günnemann**[1], **Aleksandar Bojchevski**[2]
[1]Dept. of Computer Science & Munich Data Science Institute, Technical University of Munich
[2]CISPA Helmholtz Center for Information Security
{f.mujkanovic, s.geisler, s.guennemann}@tum.de | bojchevski@cispa.de

## Abstract

A cursory reading of the literature suggests that we have made a lot of progress in designing effective adversarial defenses for Graph Neural Networks (GNNs). Yet, the standard methodology has a serious flaw – virtually all of the defenses are evaluated against non-adaptive attacks leading to overly optimistic robustness estimates. We perform a thorough robustness analysis of 7 of the most popular defenses spanning the entire spectrum of strategies, i.e., aimed at improving the graph, the architecture, or the training. The results are sobering – most defenses show no or only marginal improvement compared to an undefended baseline. We advocate using custom adaptive attacks as a gold standard and we outline the lessons we learned from successfully designing such attacks. Moreover, our diverse collection of perturbed graphs forms a (black-box) unit test offering a first glance at a model's robustness.[1]

## 1 Introduction

The vision community learned a bitter lesson – we need specific carefully crafted attacks to properly evaluate the adversarial robustness of a defense. Consequently, adaptive attacks are considered the gold standard [44]. This was not always the case; until recently, most defenses were tested only against relatively weak static attacks. The turning point was Carlini & Wagner [3]'s work showing that 10 methods for detecting adversarial attacks can be easily circumvented. Shortly after, Athalye et al. [1] showed that 7 out of the 9 defenses they studied can be broken since they (implicitly) rely on obfuscated gradients. So far, this bitter lesson is completely ignored in the graph domain.

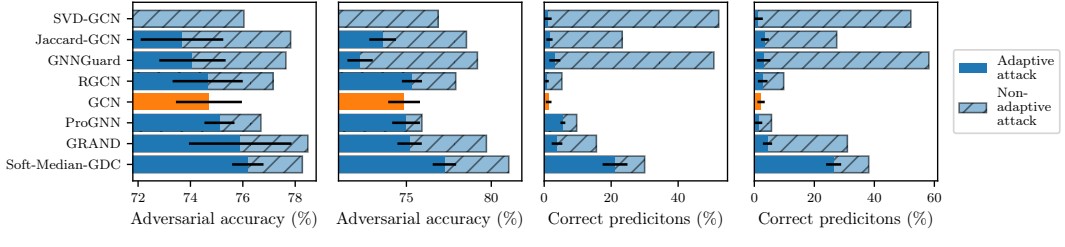

(a) Global, Poisoning    (b) Global, Evasion    (c) Local, Poisoning    (d) Local, Evasion

Figure 1: Adaptive attacks draw a different picture of robustness. All defenses are less robust than reported, with an undefended GCN [33] outperforming some. We show results on Cora ML for both poisoning (attack before training) and evasion (attack after training), and both global (attack the test set jointly) and local (attack individual nodes) setting. The perturbation budget is relative w.r.t. the #edges for global attacks (5% evasion, 2.5% poisoning) and w.r.t. the degree for local attacks (100%). In (a)/(b) SVD-GCN is catastrophically broken – our adaptive attacks reach 24%/9% (not visible). Note that our non-adaptive attacks are already stronger than what is typically used (see § 5).

---

*equal contribution    [1] Project page: https://www.cs.cit.tum.de/daml/are-gnn-defenses-robust/

36th Conference on Neural Information Processing Systems (NeurIPS 2022).

Virtually no existing work that proposes an allegedly robust Graph Neural Network (GNN) evaluates against adaptive attacks, leading to overly optimistic robustness estimates. To show the seriousness of this methodological flaw we categorize 49 works that propose a robust GNN and are published at major conferences/journals. We then choose one defense per category (usually the most highly cited). Not surprisingly, we show that none of the assessed models are as robust as originally advertised in their respective papers. In Fig. 1 we summarize the results for 7 of the most popular defenses, spanning the entire spectrum of strategies (i.e., aimed at improving the graph, the architecture, or the training, see Table 1).

We see that in both local and global settings, as well as for both evasion and poisoning, the adversarial accuracy under our adaptive attacks is significantly smaller compared to the routinely used non-adaptive attacks. Even more troubling is that many of the defenses perform worse than an undefended baseline (a vanilla GCN [33]). Importantly, the 7 defenses are not cherry-picked. We report the results for each defense we assessed and selected each defence before running any experiments.

Adversarial robustness measures the local generalization capabilities of a model, i.e., sensitivity to (bounded) worst-case perturbations. Certificates typically provide a lower bound on the actual robustness while attacks provide an upper bound. Since stronger attacks directly translate into tighter bounds our goal is to design the strongest attack possible. Our adaptive attacks have perfect knowledge of the model, the parameters, and the data, including all defensive measures. In contrast, non-adaptive attacks (e.g., transferred from an undefended proxy or an attack lacking knowledge about defense measures) only show how good the defense is at suppressing a narrow subset of input perturbations.[2]

Tramer et al. [44] showed that even adaptive attacks can be tricky to design with many subtle challenges. The graph domain comes with additional challenges since graphs are typically sparse and discrete and the representation of any node depends on its neighborhood. For this reason, we describe the recurring themes, the lessons learned, and our systematic methodology for designing strong adaptive attacks for all examined models. Additionally, we find that defenses are *sometimes* sensitive to a common attack vector and transferring attacks can also be successful. Thus, the diverse collection of perturbed adjacency matrices resulting from our attacks forms a (black-box) unit test that any truly robust model should pass before moving on to adaptive evaluation. In summary:

- We survey and categorize *49 defenses* published across prestigious machine learning venues.
- We design custom attacks for 7 defenses (14%), covering the spectrum of defense techniques. All examined models forfeit a large fraction of previously reported robustness gains.
- We provide a transparent methodology and guidelines for designing strong adaptive attacks.
- Our collection of perturbed graphs can serve as a robustness unit test for GNNs.

## 2   Background and preliminaries

We follow the most common setup and assume GNN [20, 33] classifiers $f_\theta(\mathbf{A}, \mathbf{X})$ that operate on a symmetric binary adjacency matrix $\mathbf{A} \in \{0,1\}^{n \times n}$ with binary node features $\mathbf{X} \in \{0,1\}^{n \times d}$ and node labels $\mathbf{y} \in \{1, 2, \ldots, C\}^n$ where $C$ is the number of classes, $n$ is the number of nodes, and $m$ the number of edges. A poisoning attack perturbs the graph (flips edges) prior to training, optimizing

$$\max_{\tilde{\mathbf{A}} \in \Phi(\mathbf{A})} \ell_{\text{attack}}(f_{\theta^*}(\tilde{\mathbf{A}}, \mathbf{X}), \mathbf{y}) \quad \text{s.t.} \quad \theta^* = \arg\min_\theta \ell_{\text{train}}(f_\theta(\tilde{\mathbf{A}}, \mathbf{X}), \mathbf{y}) \tag{1}$$

where $\ell_{\text{attack}}$ is the attacker's loss, which is possibly different from $\ell_{\text{train}}$ (see § 4). In an evasion attack, $\theta^*$ is kept fixed and obtained by training on the clean graph $\min_\theta \ell_{\text{train}}(f_\theta(\mathbf{A}, \mathbf{X}), \mathbf{y})$. In both cases, the locality constraint $\Phi(\mathbf{A})$ enforces a budget $\Delta$ by limiting the perturbation to an $L_0$-ball around the clean adjacency matrix: $\|\tilde{\mathbf{A}} - \mathbf{A}\|_0 \le 2\Delta$. Attacks on $\mathbf{X}$ also exist, however, this scenario is not considered by the vast majority of defenses. For example, only one out of the seven examined ones also discusses feature perturbations. We refer to § D for more details on adaptive feature attacks.

**Threat model.** Our attacks aim to either cause misclassification of the entire test set (*global*) or a single node (*local*). To obtain the strongest attack possible (i.e., tightest robustness upper bound), we use white-box attacks. We do not constrain the attacker beyond a simple budget constraint that enforces a maximum number of perturbed edges. For our considerations on unnoticeability, see § A.

---

[2]   From a security perspective non-adaptive attacks (typically transfer attacks) are also relevant since a real-world adversary is unlikely to know everything about the model and the data.

**Greedy attacks.** Attacking a GNN typically corresponds to solving a constrained discrete non-convex optimization problem that – evident by this work – is hard to solve. Commonly, approximate algorithms are used to to tackle these optimization problems. For example, the single-step Fast Gradient Attack (FGA) flips the edges whose gradient (i.e., $\nabla_{\mathbf{A}} \ell_{\text{train}}(f_{\theta^*}(\mathbf{A}, \mathbf{X}), \mathbf{y})$) most strongly indicates so. On the other hand, Nettack [67] and Metattack [66] are greedy multi-step attacks. The greedy approaches have the nice side-effect that an attack for a high budget $\Delta$ directly gives all attacks for budgets lower than $\Delta$. On the other hand, they tend to be relatively weaker.

**Projected Gradient Descent (PGD).** Alternatively, PGD [53] has been applied to GNNs where the discrete adjacency matrix is relaxed to $[0, 1]^{n \times n}$ during the gradient-based optimization and the resulting weighted change reflects the probability of flipping an edge. After each gradient update, the changes are projected back such that the budget holds in expectation $\|\mathbb{E}[\tilde{\mathbf{A}}] - \mathbf{A}\|_0 \leq 2\Delta$. Finally, multiple samples are obtained and the strongest perturbation $\tilde{\mathbf{A}}$ is chosen that obeys the budget $\Delta$. The biggest caveats while applying $L_0$-PGD are the relaxation gap and limited scalability (see Geisler et al. [17] for a detailed discussion and a scalable alternative).

**Evasion vs. poisoning.** Evasion can be considered the easier setting from an attack perspective since the model is fixed $f_{\theta^*}$. For poisoning, on the other hand, the adjacency matrix is perturbed before training (Eq. 1). Two general strategies exist for poisoning attacks: (1) transfer a perturbed adjacency matrix from an evasion attack [67]; or (2) attack directly by, e.g., unrolling the training procedure to obtain gradients through training [66]. Xu et al. [53] propose to solve Eq. 1 with alternating optimization which was shown to be even weaker than the evasion transfer (1). Note that evasion is particularly of interest for inductive learning and poisoning for transductive learning.

## 3 Adversarial defenses

We select the defenses s.t. we capture the entire spectrum of methods improving robustness against structure perturbations. For the selection, we extend the taxonomy proposed in [21]. We selected the subset without cherry-picking based on the criteria elaborated below before experimentation.

**Taxonomy.** The top-level categories are *improving the graph* (e.g., preprocessing), *improving the training* (e.g., adversarial training or augmentations), and *improving the architecture*. Many defenses for structure perturbations either fall into the category of improving the graph or adaptively weighting down edges through an improved architecture. Thus, we introduce further subcategories. Similar to [21]'s discussion, unsupervised improvement of the graph finds clues in the node features and graph structure, while supervised improvement incorporates gradient information from the learning objective. Conversely, for adaptive edge weighting, we identify three prevalent approaches: rule-based (e.g., using a simple metric), probabilistic (e.g., modeling a latent distribution), and robust aggregations (e.g., with guarantees). We assign each defense to the most fitting taxon (details in § B).

**Selected defenses.** To evaluate a diverse set of defenses, we select one per leaf taxon.[3] We prioritize highly cited defenses published at renowned venues with publicly available code. We implement all defenses in one unified pipeline. We present the categorization of defenses and our selection in Table 1. Similarly to Tramer et al. [44], we exclude defenses in the "robust training" category (see § C for a discussion). Two of the three models in the "miscellaneous" category report some improvement in robustness, but they are not explicitly designed for defense purposes so we exclude them from our study. Some works evaluate only against evasion [48], others only poisoning [12, 15, 58], and the rest tackle both [17, 30, 63]. In some cases the evaluation setting is not explicitly stated and inferred by us. For completeness, we consider each defense in all four settings (local/global and evasion/poisoning). Next, we provide a short summary of the key ideas behind each defense (details in § E).

**Improving the graph.** The feature-based *Jaccard-GCN* [48] uses a preprocessing step to remove all edges between nodes whose features exhibit a Jaccard similarity below a certain threshold. This was motivated by the homophily assumption which is violated by prior attacks that tend to insert edges between dissimilar nodes. The structure-based *SVD-GCN* [12] replaces the adjacency matrix with a low-rank approximation prior to plugging it into a regular GNN. This defense was motivated by the observation that the perturbations from Nettack tend to disproportionately affect the high-frequency spectrum of the adjacency matrix. The key idea in *ProGNN* [30] is to learn the graph structure by

---

[3] The only exception is unsupervised graph improvement, as it contains two of the most popular approaches, which rely on orthogonal principles. One filters edges based on the node features [48], the other uses a low-rank approximation of the adjacency matrix [12].

Table 1: Categorization of selected defenses. Our taxonomy extends the one by Günnemann [21].

| Taxonomy | | | Selected Defenses | Other Defenses |
|---|---|---|---|---|
| Improving graph | Unsupervised | | Jaccard-GCN [48] SVD-GCN [12] | [10, 26, 50, 59, 60] |
| | Supervised | | ProGNN [30] | [51, 43, 56] |
| Improving training | Robust training | | n/a (see § C) | [6, 9, 14, 22, 27, 28, 41, 52, 53, 54] |
| | Further training principles | | GRAND [15] | [5, 11, 29, 39, 42, 55, 61, 64, 65] |
| Improving architecture | Adaptively weighting edges | Rule-based | GNNGuard [58] | [31, 36, 37, 57] |
| | | Probabilistic | RGCN [63] | [8, 13, 24, 25, 38] |
| | | Robust agg. | Soft-Median-GDC [17] | [7, 16, 47] |
| | Miscellaneous | | n/a (see above) | [40, 46, 49] |

alternatingly optimizing the parameters of the GNN and the adjacency matrix (the edge weights). The loss for the latter includes the standard cross-entropy loss, the distance to the original graph, and three other objectives designed to promote sparsity, low rank, and feature smoothness.

**Improving the training.** *GRAND* [15] relies on random feature augmentations (zeroing features) coupled with neighbourhood augmentations $\bar{\mathbf{X}} = (\mathbf{AX} + \mathbf{AAX} + \cdots)$. All randomly augmented copies of $\bar{\mathbf{X}}$ are passed through the same MLP that is trained with a consistency regularization loss.

**Improving the architecture.** *GNNGuard* [58] filters edges in each message passing aggregation via cosine-similarity (smoothed over layers). In the first layer of *RGCN* [63] we learn a Gaussian distribution over the feature matrix and the subsequent layers then manipulate this distribution (instead of using point estimates). For the loss we then sample from the resulting distribution. In addition, in each layer, RGCN assigns higher/lower weights to features with low/high variance. *Soft-Median-GDC* [17] replaces the message passing aggregation function in GNNs (typically a weighted mean) with a more robust alternative by relaxing the median using differentiable sorting.

**Common themes.** One theme shared by some defenses is to first discover some property that can discriminate clean from adversarial edges (e.g., high vs. low feature similarity), and then propose a strategy based on that property (e.g., filter low similarity edges). Often they analyze the edges from only a single attack such as Nettack [67]. The obvious pitfall of this strategy is that the attacker can easily adapt by restricting the adversarial search space to edges that will bypass the defense's (implicit) filter. Another theme is to add additional loss terms to promote some robustness objectives. Similarly, the attacker can incorporate the same terms in the attack loss to negate their influence.

## 4 Methodology: How to design strong adaptive attacks

In this section, we describe our general methodology and the lessons we learned while designing adaptive attacks. We hope these guidelines can serve as a reference for testing new defenses.

**Step 1 – Understand how the defense works** and categorize it. For example, some defenses rely on preprocessing which filters out edges that meet certain criteria (e.g., Jaccard-GCN [48]). Others introduce additional losses during training (e.g., GRAND [15]) or change the architecture (e.g., RGCN [63]). Different defenses might need different attacks or impose extra requirements on them.

**Step 2 – Probe for obvious weaknesses.** Some examples include: (a) transfer adversarial edges from another (closely related) model (see also § 6); (b) use a gradient-free (black-box) attack. For example, in our local experiments, we use a *Greedy Brute Force* attack: in each step, it considers all possible single edge flips and chooses the one that contributes most to the attack objective (details in § A).

**Step 3 – Launch a gradient-based adaptive attack.** For rapid prototyping, use a comparably cheap attack such as FGA, and later advance to stronger attacks like PGD. For poisoning, strongly consider meta-gradient-based attacks like Metattack [66] that unroll the training procedure, as they almost always outperform just transferring perturbations from evasion. Unsurprisingly, we find that applying PGD [53] on the meta gradients often yields even stronger attacks than the greedy Metattack, and we refer to this new attack as *Meta-PGD* (details in § A).

**Step 4 – Address gradient issues.** Some defenses contain components that are non-differentiable, lead to exploding or vanishing gradients, or obfuscate the gradients [1]. To circumvent these issues, potentially: (a) adjust the defense's hyperparameters to retain numerical stability; (b) replace the offending component with a differentiable or stable counterpart, e.g., substitute the low-rank approximation of SVD-GCN [12] with a suitable differentiable alternative; or (c) remove components, e.g., drop the "hard" filtering of edges done in the preprocessing of Soft-Median-GDC [17]. These considerations also include poisoning attacks, where one also needs to pay attention to all components of the training procedure. For example, we ignore the nuclear norm loss term in the training of ProGNN [30] to obtain the meta-gradient. Of course, keep the entire defense intact for its final evaluation on the found perturbations.

**Step 5 – Adjust the attack loss.** In previous works, the attack loss is often chosen to be the same as the training loss, i.e., the cross-entropy (CE). This is suboptimal since CE is not *consistent* according to the definition by Tramer et al. [44] – higher loss values do not indicate a stronger attack. Thus, we use a variant of the consistent Carlini-Wagner loss [4] for *local* attacks, namely the logit margin (LM), i.e., the logit difference between the ground truth class and most-likely non-true class. However, as discussed by Geisler et al. [17], for *global* attacks the mean LM across all target nodes is still suboptimal since it can "waste" budget on already misclassified nodes. Their tanh logit margin (TLM) loss resolves this issue. If not indicated otherwise, we either use TLM or the probability margin (PM) loss – a slight variant of LM that computes the margin after the softmax rather than before.

**Step 6 – Tune the attack hyperparameters** such as the number of PGD steps, the attack learning rate, the optimizer, etc. For example, for Metattack we observed that using the Adam optimizer [32] can weaken the attack and replacing it with SGD can increase the effectiveness.

**Lessons learned.** We provide a detailed description of each adaptive attack and the necessary actions to make it as strong as possible in § E. Here, we highlight some important recurring challenges that should be kept in mind when designing adaptive attacks. (1) Numerical issues, e.g., due to division by tiny numbers can lead to weak attacks, and we typically resolve them via clamping. (2) In some cases we observed that for PGD attacks it is beneficial to clip the gradients to stabilize the adversarial optimization. (3) For a strong attack it is essential to tune its hyperparameters. (4) Relaxing non-differentiable components and deactivating operations that filter edges/embeddings based on a threshold in order to obtain gradients for every edge is an effective strategy. (5) If the success of evasion-poisoning transfer depends on a fixed random initialization (see § J), it helps to use multiple clean auxiliary models trained with different random seeds for the PGD attack – in each PGD step we choose one model randomly. (6) Components that make the optimization more difficult but barely help the defense can be safely deactivated. (7) It is sometimes beneficial to control the randomness in the training loop of Meta-PGD. (8) For Meta-PGD it can help to initialize the attack with non-zero perturbations and e.g., use the perturbed graph of a different attack.

**Example 1 – SVD-GCN.** To illustrate the attack process (especially steps 3 and 4) we present a case study of how we construct an adaptive attack against SVD-GCN. Gradient-free attacks like Nettack do not work well here as they waste budget on adversarial edges which are filtered out by the low-rank approximation (LRA). Moreover, to the demise of gradient-based attacks, the gradients of the adjacency matrix are very unstable due to the SVD and thus less useful. Still, we start with a gradient-based attack as it is easier to adapt, specifically FGA, whose quick runtime enables rapid prototyping as it requires only a single gradient calculation. To replace the LRA with a function whose gradients are better behaved, we first decompose the perturbed adjacency matrix $\tilde{\mathbf{A}} = \mathbf{A} + \delta\mathbf{A}$ and, thus, only need gradients for $\delta\mathbf{A}$. Next, we notice that the eigenvectors of $\mathbf{A}$ usually have few large components. Perturbations along those principal dimensions are representable by the eigenvectors, hence most likely are neither filtered out nor impact the eigenvectors. Knowing this, we approximate the LRA in a tractable manner by element-wise multiplication of $\delta\mathbf{A}$ with weights that quantify how well an edge aligns with the principal dimensions (details in § E). In short we replace $\mathrm{LRA}(\mathbf{A} + \delta\mathbf{A})$ with $\mathrm{LRA}(\mathbf{A}) + \delta\mathbf{A} \circ \mathrm{Weight}(\mathbf{A})$, which admits useful gradients. This approach carries over to other attacks such as Nettack – we can incorporate the weights into its score function to avoid selecting edges that will be filtered out.

**Example 2 – ProGNN.** While we approached SVD-GCN with a theoretical insight, breaking a composite defense like ProGNN requires engineering and tinkering. When attacking ProGNN with PGD and transferring the perturbations to poisoning we observe that the perturbations are only effective if the model is trained with the same random seed. This over-sensitivity can be avoided by

employing lesson (5) in § 4. As ProGNN is very expensive to train due to its nuclear norm regularizer, we drop that term when training the set of auxiliary models without hurting attack strength. For unrolling the training we again drop the nuclear norm regularizer since it is non-differentiable. Sometimes PGD does not find a state with high attack loss, which can be alleviated by random restarts. As Meta-PGD optimization quickly stalls, we initialize it with a strong perturbation found by Meta-PGD on GCN. All of these tricks combined are necessary to successfully attack ProGNN.

**Effort.** Breaking Jaccard-GCN (and SVD-GCN) required around half an hour (resp. three days) of work for the initial proof of concept. Some other defenses require various adjustments that need to be developed over time, but reusing those can quickly break even challenging defenses. It is difficult to quantify this effort, but it can be greatly accelerated by adopting our lessons learned in § 4. In any case, we argue that authors proposing a new defense must put in reasonable effort to break it.

## 5    Evaluation of adaptive attacks

First, we provide details on the experimental setup and used metrics. We then report the main results and findings. We refer to § A for details on the base attacks, including our Greedy Brute Force and Meta-PGD approaches. We provide the code, configurations, and a collection of perturbed graphs on the project website linked on the first page.

**Setup.** We use the two most widely used datasets in the literature, namely Cora ML [2] and Citeseer [19] (details in § F). Unfortunately, larger datasets are barely possible since most defenses are not very scalable. Still, in § N, we discuss scalability and apply an adaptive attack to arXiv (170k nodes) [23]. We repeat the experiments for five different data splits (10% training, 10% validation, 80% testing) and report the means and variances. We use an internal cluster with Nvidia GTX 1080Ti GPUs. Most experiments can be reproduced within a few hours. However, the experiments with ProGNN and GRAND will likely require several GPU days.

**Defense hyperparameters.** When first attacking the defenses, we observed that many exhibit poor robustness using the hyperparameters provided by their authors. To not accidentally dismiss a defense as non-robust, we tune the hyperparameters such that the clean accuracy remains constant but the robustness w.r.t. adaptive attacks is improved. Still, we run all experiments on the untuned defenses as well to confirm we achieve this goal. In the same way, we also tune the GCN model, which we use as a reference to asses whether a defense has merit. We report the configurations and verify the success of our tuning in § H.

**Attacks and budget.** In the *global* setting, we run the experiments for budgets $\Delta$ of up to 15% of the total number of edges in the dataset. Due to our (R)AUC metric (see below), we effectively focus on only the lower range of evaluated budgets. We apply FGA and PGD [53] for evasion. For poisoning, we transfer the found perturbations and also run Metattack [66] and our Meta-PGD. Recall that where necessary, we adapt the attacks to the defenses as outlined in § 4 and detailed in § E.

In the *local* setting, we first draw sets of 20 target nodes per split with degrees 1, 2, 3, 5, 8-10, and 15-25 respectively (total of 120 nodes). This enables us to study how the attacks affect different types of nodes – lower degree nodes are often conjectured to be less robust (see also § K). We then run the experiments for relative budgets $\Delta$ of up to 200% of the target node's degree. For example, if a node has 10 neighbors, and the budget $\Delta = 70\%$ then the attacker can change up to $10 \cdot 0.7 = 7$ edges. This commonly used setup ensures that we treat both low and high-degree nodes fairly. We use Nettack [67], FGA, PGD, and our greedy brute force attack for evasion. For poisoning, we only transfer the found perturbations. Again, we adapt the attacks to the defenses if necessary.

In alignment with our threat model, we evaluate each found perturbation by the test set accuracy it achieves (*global*) or the ratio of target nodes that remain correctly classified (*local*). For each budget, we choose the strongest attack among all attempts (e.g., PGD, Metattack, Meta-PGD). This gives rise to an envelope curve as seen in Fig. 3. We also include lower budgets as attempts, i.e., we enforce the envelope curve to be monotonically decreasing.

We introduce a rich set of attack characteristics by also transferring the perturbations supporting the envelope curve to every other defense. These transfer attacks then also contribute to the final envelope curve of each defense, but in most cases their contribution is marginal.

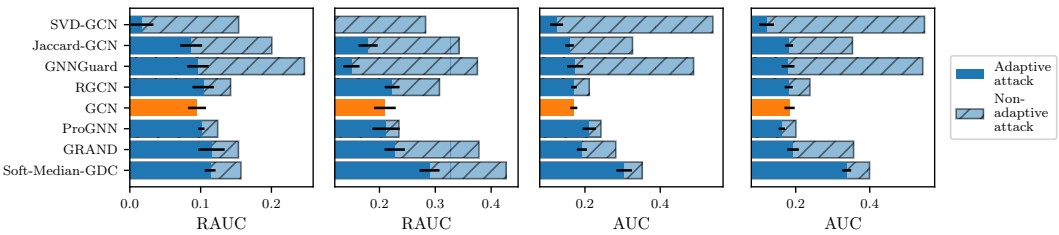

(a) Global, Poisoning  (b) Global, Evasion  (c) Local, Poisoning  (d) Local, Evasion

Figure 2: Adaptive vs. non-adaptive attacks with budget-agnostic (R)AUC on Cora ML (c.f. Fig. 1). SVD-GCN (b) is disastrously broken – our adaptive attacks reach <0.02 (not visible). § F for Citeseer.

**Non-adaptive attacks.** We call any attack "non-adaptive" that is not aware of any changes made to the model (including defense mechanisms). Where we report results for a non-adaptive attack (e.g., Fig. 1 or Fig. 2), we specifically refer to an attack performed on a (potentially linearlized) GCN with commonly used hyperparameters (i.e., untuned). We then apply the perturbed adjacency matrix to the actual defense. In other words, we transfer the adversarial perturbation from a GCN. For our *local* non-adaptive attack, we always use Nettack. In contrast, for our *global* non-adaptive attack, we apply all attacks listed above, and then transfer for each budget the attack which is strongest against the GCN. Due to this ensemble of attacks, our global non-adaptive attack is expected to be slightly stronger than the non-adaptive attacks in most other works.

**Area Under the Curve (AUC).** An envelope curve gives us a detailed breakdown of the empirical robustness of a defense for different adversarial budgets. However, it is difficult to compare different attacks and defenses by only visually comparing their curves in a figure (e.g., see Fig. 4). Therefore, in addition to this breakdown per budget, we summarize robustness using the Area Under the Curve (AUC), which is independent of a specific choice of budget $\Delta$ and also punishes defenses that achieve robustness by trading in too much clean accuracy. Intuitively higher AUCs indicate more robust models, and conversely, lower AUCs indicate stronger attacks.

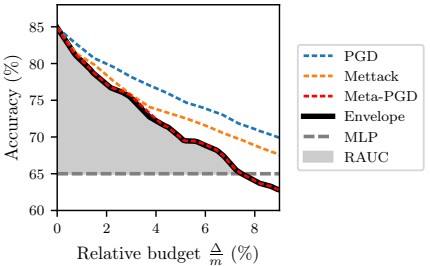

Figure 3: The dotted lines show the test set accuracy per budget after three global poisoning attacks against a tuned GCN on Cora ML. Taking the envelope gives the solid black robustness curve. The dashed gray line denotes the accuracy of an MLP. The shaded area is the RAUC.

As our *local* attacks break virtually all target nodes within our conservative maximum budget (see § F), taking the AUC over all budgets conveniently measures how quick this occurs. However, for *global* attacks, the test set accuracy continues to decrease for unreasonably large budget, and it is unclear when to stop. To avoid having to choose a maximum budget, we wish to stop when discarding the entire tainted graph becomes the better defense. This is fulfilled by the area between the envelope curve and the line signifying the accuracy of an MLP – a model that is oblivious to the graph structure, at the expense of a substantially lower clean accuracy than a GNN. We call this metric Relative AUC (RAUC) and illustrate it in Fig. 3. More formally, $\mathrm{RAUC}(c) = \int_0^{b_0} (c(b) - a_{\mathrm{MLP}})\mathrm{d}b$ s.t. $b \lessgtr b_0 \implies c(b) \gtrless a_{\mathrm{MLP}}$ where $c(\cdot)$ is a piecewise linear robustness per budget curve, and $a_{\mathrm{MLP}}$ is the accuracy of the MLP baseline. We normalize the RAUC s.t. 0% is the performance of an MLP and 100% is the optimal score (i.e., 100% accuracy).

**Finding 1 – Our adaptive attacks lower robustness by 40% on average.** In Fig. 2 we compare non-adaptive attacks, the current standard to evaluate defenses, with our adaptive attacks which we propose as a new standard. The achieved (R)AUC in each case drops on average by 40% (similarly for Citeseer, see § F). In other words, the reported robustness in the original works proposing a defense is roughly 40% too optimistic. We confirm a statistically significant drop ($p < 0.05$) with a one-sided t-test in 85% of all cases. Considering adversarial accuracy for (small) fixed adversarial budget (Fig. 1) instead of the summary (R)AUC over all budgets tells the same story: non-adaptive attacks are too weak to be reliable indicators of robustness and adaptive attacks massively shrink the alleged robustness gains.

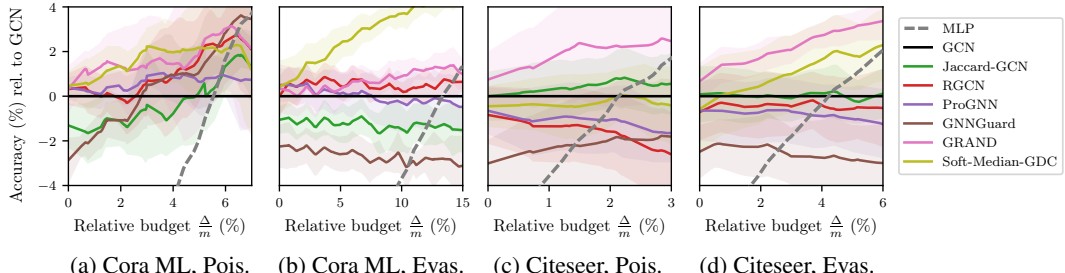

(a) Cora ML, Pois.   (b) Cora ML, Evas.   (c) Citeseer, Pois.   (d) Citeseer, Evas.

Figure 4: Difference (defense – undefended GCN) of adversarial accuracy for the strongest global attack per budget. Almost half of the defenses perform worse than the GCN. We exclude SVD-GCN since it is catastrophically broken and plotting it would make the other defenses illegible (accuracy <24% already for a budget of 2% on Cora ML). Absolute numbers in § F.

**Finding 2 – Structural robustness of GCN is not easily improved.** In Fig. 4 (global) and Fig. 5 (local) we provide a more detailed view for different adversarial budgets and different graphs. For easier comparison we show the accuracy relative to the undefended GCN baseline. Overall, the decline is substantial. Almost half of the examined defenses perform worse than GCN and most remaining defenses neither meaningfully improve nor lower the robustness (see also Fig. 1 and Fig. 3). GRAND and Soft-Medoid-GCN retain robustness in some settings, but the gains are smaller than reported.

**Finding 3 – Defense effectiveness depends on dataset.** As we can see in Fig. 4 and Fig. 5, our ability to circumvent specific defenses tends to depend on the dataset. It appears that some defenses are more suited for different datasets. For example, GRAND seems to be a good choice for Citeseer while it is not as strong on Cora ML. The results for local attacks (Fig. 5) paint a similar picture, here we see that Cora ML is more difficult to defend. This points to another potentially problematic pitfall: most defenses are developed only using these two datasets as benchmarks. Is robustness even worse on other graphs? We leave this question for future work.

**Finding 4 – No trade-off between accuracy and robustness for structure perturbations.** Instead, Fig. 6 shows that defenses with high clean accuracy also exhibit high RAUC, i.e., are more robust against our attacks. This appears to be in contrast to the image domain [45]. However, we cannot exclude that future more powerful defenses might manifest this trade-off in the graph domain.

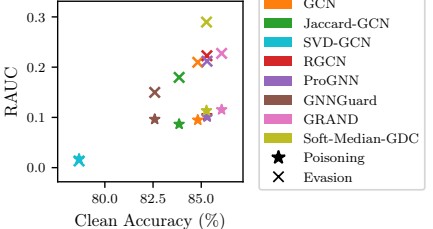

Figure 6: Model accuracy vs. RAUC of the strongest global attacks on Cora ML. We do not observe a robustness accuracy trade-off, but even find models with higher accuracy to be more robust.

**Finding 5 – Necessity of adaptive attacks.** In Fig. 7, we show two exemplary characteristics of how an adaptive attack bypasses defensive measures. First, to attack SVD-GCN, it seems particularly effective to insert connections to high-degree nodes. Second, for GNNGuard, GRAND and Soft-Median-GDC it is disproportionally helpful to

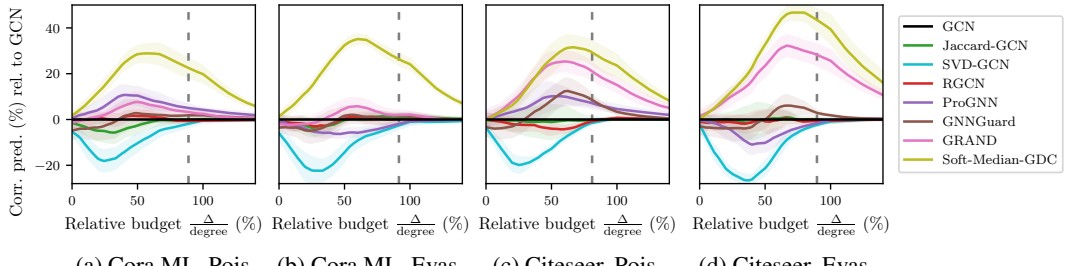

(a) Cora ML, Pois.   (b) Cora ML, Evas.   (c) Citeseer, Pois.   (d) Citeseer, Evas.

Figure 5: Difference (defense – undefended GCN) of fraction of correct predictions for the strongest local attack per budget. Most defenses show no or only marginal gain in robustness. The dashed vertical line shows where 95% of nodes for a GCN are misclassified on average. Abs. numbers in § F.

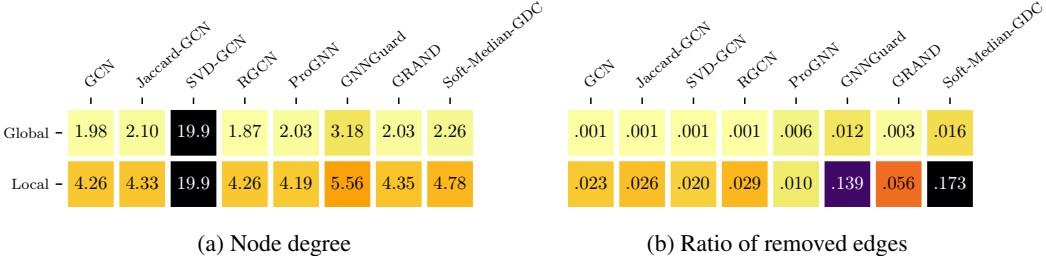

(a) Node degree                    (b) Ratio of removed edges

Figure 7: Exemplary metrics characterizing the attack vector our strongest attacks, which are those visible in Fig. I.1 and Fig. I.2. We give a more elaborate study of attack characteristics in § L.

delete edges. These examples illustrate why the existence of a one-fits-all perturbation which circumvents all possible defenses is unlikely. Instead, an adaptive attack is necessary to properly assess a defense's efficacy since different models are particularly susceptible to different perturbations.

**Additional analysis.** During this project, we generated a treasure trove of data. We perform a more in-depth analysis of our attacks in the appendix. First, we study how node degree affects attacks (see § K). For local attacks, the required budget to misclassify a node is usually proportional to the node's degree. Global attacks tend to be oblivious to degree and uniformly break nodes. Next, we perform a breakdown of each defense in terms of the sensitivity to different attacks (see § I). In short, global attacks are dominated by PGD for evasion and Metattack/Meta-PGD for poisoning with the PM or TLM loss. For local, our greedy brute-force is most effective, rarely beaten by PGD and Nettack. Finally, we analyze the properties of the adversarial edges in terms of various graph statistics such as edge centrality and frequency spectra (see § L § M).

## 6 Robustness unit test

Next we systematically study how well the attacks transfer between defenses, as introduced in the *attacks and budget* paragraph in § 5. In Fig. 8, we see that in 15 out of 16 cases the adaptive attack is the most effective strategy (see main diagonal). However for many defenses, there is often a source model or ensemble of source models (for the latter see § G) which forms a strong transfer attack.

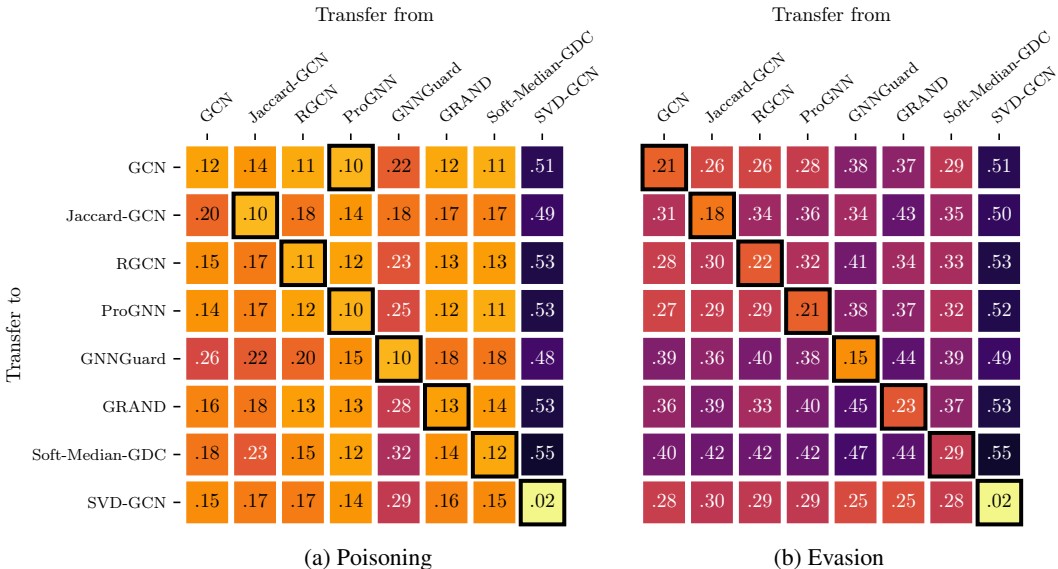

(a) Poisoning                    (b) Evasion

Figure 8: RAUC for the transfer of the strongest global adaptive attacks on Cora ML between models. The columns contain the models for which the adaptive attacks were created. The rows contain the RAUC after the transfer. With only one exception, adaptive attacks (diagonal) are most effective.

Motivated by the effectiveness of transfer attacks (especially if transferring from ProGNN [30]), we suggest this set of perturbed graphs to be used as a bare minimum robustness unit test: one can probe a new defense by testing against these perturbed graphs, and if there exists at least one that diminishes the robustness gains, we can immediately conclude that the defense is not robust in the worst-case – without the potentially elaborate process of designing a new adaptive attack. We provide instructions on how to use this collection in the accompanying code.

Nevertheless, we cannot stress enough that this collection does not replace a properly developed adaptive attack. For example, if one would come up with SVD-GCN and would use our collection (excluding the perturbed graphs for SVD-GCN) the unit test would partially pass. However, as we can see in e.g., Fig. 2, SVD-GCN can be broken with an – admittedly very distinct – adaptive attack.

## 7 Related work

Excluding attacks on undefended GNNs, previous works studying adaptive attacks in the graph domain are scarce. The recently proposed graph robustness benchmark [62] also only studies transfer attacks. Such transfer attacks are so common in the graph domain that their usage is often not even explicitly stated, and we find that the perturbations are most commonly transferred from Nettack or Metattack (both use a linearized GCN). Other times, the authors of a defense only state that they use PGD [53] (aka "topology attack") without further explanations. In this case, the authors most certainly refer to a PGD transfer attack on a GCN proxy. They almost never apply PGD to their actual defense, which would yield an adaptive attack (but possibly weak, see § 4 for guidance).

An exception where the defense authors study an adaptive attack is SVD-GCN [12]. Their attack collects the edges flipped by Nettack in a difference matrix $\delta\mathbf{A}$, replaces its most significant singular values and vectors with those from the clean adajcency matrix $\mathbf{A}$, and finally adds it to $\mathbf{A}$. Notably, this yields a dense continuous perturbed adjacency matrix. While their SVD-GCN is susceptible to these perturbations, the results however do not appear as catastrophic as with our adaptive attacks, despite their severe violation of our threat model (see § 2). Geisler et al. [17] are another exception where gradient-based greedy and PGD attacks are directly applied to their Soft-Median-GDC defense, making them adaptive. Still, our attacks manage to further reduce their robustness estimate.

## 8 Discussion

We hope that the adversarial learning community for GNNs will reflect on the bitter lesson that evaluating adversarial robustness is not trivial. We show that on average adversarial robustness estimates are overstated by 40%. To ease the transition into a more reliable regime of robustness evaluation for GNNs we share our recipe for successfully designing strong adaptive attacks.

Using adaptive (white-box) attacks is also interesting from a security perspective. If a model successfully defends such strong attacks, it is less likely to have remaining attack vectors for a real-world adversary. Practitioners can use our methodology to evaluate their models in hope to avoid an arms race with attackers. Moreover, the white-box assumption lowers the chance that real-world adversaries can leverage our findings, as it is unlikely that they have perfect knowledge.

We also urge for caution since the attacks only provide an upper bound (which with our attacks is now 40% tighter). Nevertheless, we argue that the burden of proof that a defense is truly effective should lie with the authors proposing it. Following our methodology, the effort to design a strong adaptive attack is reduced, so we advocate for adaptive attacks as the gold-standard for future defenses.

### Acknowledgments and Disclosure of Funding

This research was supported by the Helmholtz Association under the joint research school "Munich School for Data Science – MUDS".

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
