# A Attacks overview

In this section, we make the ensemble of attacks explicit and explain essential details. We then adapt these attack primitives to circumvent the defense mechanisms (see § E).

**Global evasion attacks.** The goal of a global attack is to provoke the misclassification of a large fraction of nodes (i.e., the test set) jointly, crafting a single perturbed adjacency matrix. For evasion, we use *(1) the Fast Gradient Attack (FGA)* and *(2) Projected Gradient Descent (PGD)*. In FGA, we calculate the gradient towards the entries of the clean adjacency matrix $\nabla_{\mathbf{A}}\ell_{\text{attack}}(f_{\theta*}(\mathbf{A}, \mathbf{X}), \mathbf{y})$ and then flip the highest-ranked edges at once s.t. we exhaust the budget $\Delta$. In contrast, PGD requires multiple gradient updates since it uses gradient ascent (see § 2 or explanation below for Meta-PGD). We deviate from the PGD implementation of Xu et al. [53] is two ways: (I) we adapt the initialization of the perturbation before the first attack gradient descent step and (II) we adjust the final sampling of $\tilde{\mathbf{A}}$. See below for more details.

**Global poisoning attacks.** We either (a) transfer the perturbation $\tilde{\mathbf{A}}$ found by evasion attack (1) or (2) and use it to poison training, or (b) differentiate through the training procedure by unrolling it, thereby obtaining a meta gradient. The latter approach is taken by both *(3) Metattack* [66] and *(4) our Meta-PGD*. Metattack greedily flips a single edge in each iteration and then obtains a new meta gradient at the changed adjacency matrix. In Meta-PGD, we follow the same relaxation as Xu et al. [53] (see below as well as § 2) and obtain meta gradients at the relaxed adjacency matrices. In contrast to the greedy approach of Metattack, Meta-PGD is able to revise early decisions later on.

**Meta-PGD.** Next, we explain the details of Meta-PGD and we present the pseudo code for reference in Algorithm A.1. Recall that the discrete edges are relaxed $\{0, 1\} \rightarrow [0, 1]$ and that the "weight" of the perturbation reflects the probability of flipping the respective edge.

---

**Algorithm A.1** Meta-PGD

1: **Input:** Adjacency matrix $\mathbf{A}$, node features $\mathbf{X}$, labels $\mathbf{y}$, GNN $f_\theta(\cdot)$, loss $\ell_{\text{attack}}$
2: **Parameters:** Budget $\Delta$, iterations $E$, learning rates $\alpha_t$
3: Initialize $\mathbf{P}_0 \in \mathbb{R}^{n \times n}$
4: **for** $t \in \{1, 2, \ldots, E\}$ **do**
5:      Step $\mathbf{P}^{(t)} \leftarrow \mathbf{P}^{(t-1)} + \alpha_t \nabla_{\mathbf{P}^{(t-1)}} \left[ \ell_{\text{attack}} \left( f\big(\mathbf{A} + \mathbf{P}^{(t-1)}, \mathbf{X}; \theta = \text{train}(\mathbf{A} + \mathbf{P}^{(t-1)}, \mathbf{X}, \mathbf{y})\big), \mathbf{y}\right)\right]$
6:      Projection $\mathbf{P}^{(t)} \leftarrow \Pi_{\|\mathbb{E}[\mathbf{A}+\mathbf{P}^{(t)}] - \mathbf{A}\|_0 \leq 2\Delta}(\mathbf{P}^{(t)})$
7: Sample $\tilde{\mathbf{A}}$ s.t. $\|\tilde{\mathbf{A}} - \mathbf{A}\|_0 \leq 2\Delta$
8: Return $\tilde{\mathbf{A}}$

---

In the first step of Meta-PGD, we initialize the perturbation (line 3). In contrast to Xu et al. [53]'s suggestion, we find that initializing the perturbation with the zero matrix can cause convergence issues. Hence, we alternatively initialize the perturbation with $\tilde{\mathbf{A}}$ from an attack on a different model (see also lesson learned #8 in § 4).

In each attack iteration, a gradient ascent step is performed on the relaxed perturbed adjacency matrix $\tilde{\mathbf{A}}^{(t-1)} = \mathbf{A} + \mathbf{P}^{(t-1)}$ (line 5). For obtaining the meta gradient through the training process, the training is unrolled. For example, with vanilla gradient descent for training $f_\theta(\mathbf{A}, \mathbf{X}) = f(\mathbf{A}, \mathbf{X}; \theta)$, the meta gradient resolves to

$$\nabla_{\mathbf{P}^{(t-1)}} \left( \ell_{\text{attack}} \left[ f\big(\mathbf{A} + \mathbf{P}^{(t-1)}, \mathbf{X}; \theta = \theta_0 - \eta \sum_{k=1}^{E_{\text{train}}} \nabla_{\theta_{k-1}} \ell_{\text{train}}[f(\mathbf{A} + \mathbf{P}^{(t-1)}, \mathbf{X}; \theta = \theta_{k-1}), \mathbf{y}]\big), \mathbf{y}\right] \right) \quad \text{(A.1)}$$

with number of training epochs $E_{\text{train}}$, fixed training learning rate $\eta$, and parameters after (random) initialization $\theta_0$. Notice that to obtain our variant of non-meta PGD, it suffices to replace the gradient computation in line 5 with $\nabla_{\mathbf{P}^{(t-1)}} \left[ \ell_{\text{attack}}(f_{\theta*}(\mathbf{A} + \mathbf{P}^{(t-1)}, \mathbf{X}), \mathbf{y})\right]$.

Thereafter in line 6, the perturbation is projected such that in expectation the budget is obeyed, i.e., $\Pi_{\|\mathbb{E}[\mathbf{A}+\mathbf{P}^{(t)}] - \mathbf{A}\|_0 \leq 2\Delta}$. First, the projection clips $\mathbf{A} + \mathbf{P}^{(t-1)}$ to be in $[0, 1]$. If the budget is violated after clipping, it solves

$$\arg \min_{\hat{\mathbf{P}}^{(t)}} \|\hat{\mathbf{P}}^{(t)} - \mathbf{P}^{(t)}\|_2 \qquad \text{s.t.} \quad \mathbf{A} + \hat{\mathbf{P}}^{(t)} \in [0, 1]^{n \times n} \text{ and } \sum |\hat{\mathbf{P}}^{(t)}| \leq 2\Delta \qquad \text{(A.2)}$$

After the last iteration (line 7), each element of $\mathbf{P}^{(t)}$ is interpreted as a probability and multiple perturbations are sampled accordingly. The strongest drawn perturbed adjacency matrix (in terms of

attack loss) is chosen as $\tilde{\mathbf{A}}$. Specifically, in contrast to [53], we sample $K = 100$ potential solutions that all obey the budget $\Delta$ and then choose the one that maximizes the attack loss $\ell_{\text{attack}}$.

**Local attacks.** For local attacks we only run evasion attacks, and then transfer them to poisoning. This is common practice (e.g., see Zügner et al. [67] or Li et al. [34]). The attacks we use are *(1) FGA*, *(2) PGD*, *(3) Nettack [67]*, and a *(4) Greedy Brute Force* attack. Nettack greedily flips the best edges considering a linearized GCN, whose weights are either specially trained or taken from the attacked defense. In contrast, in each iteration, our Greedy Brute Force attack flips the current worst-case edge for the attacked model. It determines the worst-case perturbation by evaluating the model for every single edge flip. Notice that all examined models use two propagation steps, so we only consider all potential edges adjoining the target node or its neighbors[4]. Importantly, Greedy Brute Force is adaptive for any kind of model. Runtime-wise, the algorithm evaluates the attacked model $\mathcal{O}(\Delta n d)$ times with the number of nodes $n$ and the degree of the target node $d$. We provide pseudo code in Algorithm A.2.

---

**Algorithm A.2** Greedy Brute Force

---

1: **Input:** Target node $i$, adjacency matrix $\mathbf{A}$, node features $\mathbf{X}$, labels $\mathbf{y}$, GNN $f_\theta(\cdot)$, loss $\ell_{\text{attack}}$
2: **Parameter:** Budget $\Delta$
3: Initialize $\tilde{\mathbf{A}}^{(0)} = \mathbf{A}$
4: **for** $t \in \{1, 2, \ldots, \Delta\}$ **do**
5:     **for** potential edge $e$ adjoining $i$ or any of $i$'s direct neighbors **do**
6:         Flip edge $\tilde{\mathbf{A}}^{(t)} \leftarrow \tilde{\mathbf{A}}^{(t-1)} \pm e$
7:         Remember best $\tilde{\mathbf{A}}^{(t)}$ in terms of $\ell_{\text{attack}}(f_{\theta^*}(\tilde{\mathbf{A}}^{(t)}, \mathbf{X}), \mathbf{y})$
8:     **if** node $i$ is missclassifed **then**
9:         Return $\tilde{\mathbf{A}}^{(t)}$
10:    Recover best $\tilde{\mathbf{A}}^{(t)}$
11: Return $\tilde{\mathbf{A}}_\Delta$

---

**Unnoticeability** typically serves as a proxy to ensure that the label of an instance (here node) has not changed. In the image domain, it is widely accepted that a sufficiently small perturbation of the input image w.r.t. an $L_p$-norm is unnoticeable (and similarly for other threat models such as rotation). For graphs the whole subject of unnoticeability is more nuanced. The only constraint we use is the number of edge insertions/deletion, i.e., an $L_0$-ball around the clean adjacency matrix.

The only additional unnoticeability constraint proposed in the literature compares the clean and perturbed graph under a power law assumption on the node degrees [67]. However, we do not include such a constraint since (1) the degree distribution is only one (arbitrary) property to distinguish two graphs. (2) The degree distribution is a global property with an opaque relationship to the local class labels in node classification. (3) As demonstrated in Zügner & Günnemann [66], enforcing an indistinguishable degree distribution only has a negligible influence on attack efficacy, i.e., their gradient-based/adaptive attack conveniently circumvents this measure. Thus, we argue that enforcing such a constraint is similar to an additional (weak) defense measure and is not the focus of this work. Finally, since many defense (and attack) works in the literature considering node-classification (including the ones we study) also only use an $L_0$-ball constraint as a proxy for unnoticeability, we do the same for improved consistency. Out of scope are also other domains, like combinatorial optimization, where unnoticeability is not required since the true label of the perturbed instance is known [18].

---

[4] Due to GCN-like normalization (see § E), the three-hop neighbors need to be considered to be exhaustive. However, it is questionable if perturbing a neighbor three hops away is ever the strongest perturbation there is.

# B Defense taxonomy

Next, we give further details behind our reasoning on how to categorize defenses for GNNs. Our taxonomy extends and largely follows Günnemann [21]'s. The three main categories are *improving the graph* (§ B.1), *improving the training* (§ B.2), and *improving the architecture* (§ B.3). We assign each defense to the category that fits best, even though some defenses additionally include ideas fitting into other categories as well. For the assignment of defenses see Table 1.

## B.1 Improving the graph

With this category, we refer to all kinds of preprocessing of the graph. Alternatively, some approaches make the graph learnable with the goal of improved robustness. In summary, this category addresses changes that take place *prior* to the GNN (i.e., any message passing). We further distinguish *(1) unsupervised* and *(2) supervised* approaches.

**Unsupervised.** Any improvements that are not entangled with a learning objective, i.e., pure preprocessing, usually arising from clues found in the node features and graph structure. For example, Jaccard-GCN [48] filters out edges based on the Jaccard similarity of node features, while SVD-GCN [12] performs a low-rank approximation to filter out high-frequency perturbations. Most other approaches from this category exploit clues from features and structure simultaneously.

**Supervised.** These graph improvements are entangled with the learning objective by making the adjacency matrix learnable, often accompanied by additional regularization terms that introduce expert assumptions about robustness. For example, ProGNN [30] treats the adjacency matrix like a learnable parameter, and adds loss terms s.t. it remains close to the original adjacency matrix and exhibits properties which are assumed about clean graphs like low-rankness.

## B.2 Improving the training

These approaches improve training – without changing the architecture – s.t. the learned parameters $\theta^*$ of the GNN exhibit improved robustness. In effect, the new training "nudges" a regular GNN towards being more robust. We distinguish *(1) robust training* and *(2) further training principles*.

**Robust training.** Alternative training schemes and losses which reward the correct classification of synthetic adversarial perturbations of the training data. With this category, Günnemann [21] targets both straightforward adversarial training and losses stemming from certificates (i.e., improving certifiable robustness). Neither approach is interesting to us: the former is discussed in § C, and the latter targets provable robustness which does not lend itself to empirical evaluation.

**Further training principles.** This category is distinct from robust training due to the lack of a clear mathematical definition of the training objective. It mostly captures augmentations [15, 29, 39, 42, 61] or alternative training schemes [5, 11, 55, 64] that encourage robustness. A simple example for such an approach is to pre-train the GNN weights on perturbed graphs [42]. Another recurring theme is to use multiple models during training and then, e.g., enforce consistency among them [5].

## B.3 Improving the architecture

Even though there are some exceptions (see sub-category *(2) miscellaneous*), the recurring theme in this category is to somehow weight down the influence of some edges adaptively for each layer or message passing aggregation. We refer to this type of improved architecture with *(1) adaptively weighting edges*. We further distinguish between approaches that are *(a) rule-based*, *(b) probabilistic*, or use *(c) robust aggregation*.

*Rule-based* approaches typically use some metric [31, 58], alternative message passing [36, 37], or an auxiliary MLP [57] to filter out alleged adversarial edges. *Probabilistic* approaches either work with distributions in the latent space [63], are built upon probabilistic principles like Bayesian uncertainty quantification [13], or integrate sampling into the architecture and hence apply it also at inference time [8, 24, 25, 38]. *Robust aggregation* defenses replace the message passing aggregation (typically mean) with a more robust equivalent such as a trimmed mean, median, or soft median [7, 17]. In relation to the trimmed mean, in this category we include also other related approaches that come with some guarantees based on their aggregation scheme Wang et al. [47].

## C   On adversarial training defenses

The most basic form of adversarial training for structure perturbations aims to solve:

$$\min_{\theta} \max_{\mathbf{A}' \in \Phi(\mathbf{A})} \ell(f_{\theta}(\mathbf{A}', \mathbf{X}), \mathbf{y}) \tag{C.1}$$

Similarly to [44, 1, 4], we exclude defenses that build on adversarial training in our study for three reasons.

First, we observe that adversarial training requires knowing the clean $\mathbf{A}$. However, for poisoning, we would need to substitute $\mathbf{A}$ with an adversarially perturbed adjacency matrix $\tilde{\mathbf{A}}$. In this case, adversarial training aims to enforce adversarial generalization $\mathbf{A}' \in \Phi(\tilde{\mathbf{A}})$ for the adversarially perturbed adjacency matrix $\tilde{\mathbf{A}}$ – potentially even reinforcing the poisoning attack.

Second, an adaptive poisoning attack on adversarial training is very expensive as we need to unfold many adversarial attacks for a single training. Thus, designing truly adaptive poisoning attacks requires a considerable amount of resources. *Scaling* these attacks to such complicated training schemes is not the main objective of this work.

Third, adversarial training for structure perturbations on GNNs seems to be an unsolved question. So far, the robustness gains come from additional and orthogonal tricks such as self-training [53]. Hence, adversarial training for structure perturbations requires an entire paper on its own.

## D   On defenses against feature perturbations

As introduced in § 2, attacks may perturb the adjacency matrix $\mathbf{A}$, the feature matrix $\mathbf{X}$, or both. However, during our survey we found that few defenses tackle feature perturbations. Similarly, 6 out of the 7 defenses chosen by us mainly based on general popularity turn out to not consciously defend against feature perturbations.

The only exception is SVD-GCN [12], which also applies its low-rank approximation to the binary feature matrix. However, the authors do not report robustness under feature-only attacks; instead, they only consider mixed structure and feature attacks found by Nettack. Given the strong bias of Nettack towards structure perturbations, we argue that their experimental results do not confirm feature robustness. Correspondingly, in preliminary experiments we were not able to achieve considerable robustness gains of SVD-GCN compared to an undefended GCN – even with non-adaptive feature perturbations. If a non-adaptive attack is strong enough, there is not much merit in applying an adaptive attack.

To reiterate, due to the apparent scarcity of defenses apt against feature attacks, we decided to focus our efforts on structure attacks and defenses. However, new defenses considering feature perturbations should study robustness in the face of adaptive attacks – similarly to our work. In the following, we give some important hints for adaptive attacks using feature perturbations. We leave attacks that jointly consider feature and structure perturbations for future work due to the manifold open challenges, e.g., balancing structure and feature perturbations in the budget quantity.

**Baseline.** To gauge the robustness of defenses w.r.t. global attacks, we introduce the RAUC metric, which employs the accuracy of an MLP – which is perfectly robust w.r.t. structure perturbations – to determine the maximally sensible budget to include in the summary. As MLPs are however vulnerable to feature attacks, a different baseline model is required for this new setting. We propose to resolve this issue by using a label propagation approach, which is oblivious to the node features and hence perfectly robust w.r.t. feature perturbations.

**Perturbations.** The formulation of the set of admissible perturbations depends on what modality the data represents, which may differ between node features and graph edges. Convenient choices for continuous features are l-p-norms; in other cases, more complicated formulations are more appropriate. Accordingly, one has to choose an appropriate constrained optimization scheme.

# E  Examined adversarial defenses

In this section, we portray each defense and how we adapted the base attacks to each one. We refer to Table H.1 for the used hyperparameter values for each defense. We give the used attack parameters for a GCN below and refer to the provided code for the other defenses.

**GCN.** We employ an undefended GCN [33] as our baseline. A GCN first adds self loops to the adjacency matrix $\mathbf{A}$ and subsequently applies GCN-normalization, thereby obtaining $\mathbf{A}' = (\mathbf{D} + \mathbf{I})^{-\frac{1}{2}}(\mathbf{A} + \mathbf{I})(\mathbf{D} + \mathbf{I})^{-\frac{1}{2}}$ with the diagonal degree matrix $\mathbf{D} \in \mathbb{N}^{n \times n}$. Then, in each GCN layer it updates the hidden states $\mathbf{H}^{(l)} = \text{dropout}(\sigma(\mathbf{A}'\mathbf{H}^{(l-1)}\mathbf{W}^{(l-1)} + \mathbf{b}^{(l-1)}))$ where $\mathbf{H}^{(0)} = \mathbf{X}$. We use the non-linear ReLU activation for intermediate layers. Dropout is deactivated in the last layer and we refer to the output before softmax activation as logits. We use Adam [32] to learn the model's parameters.

**Attack.** We do not require special tricks since the GCN is fully differentiable and does not come with defensive measures to consider. In fact, the off-the-shelf attacks we employ are tailored to a GCN. For PGD, we use $E = 200$ iterations, $K = 100$ samples, and a base learning rate of 0.1. For Meta-PGD, we only lower the base learning rate to 0.01 and add gradient clipping to 1 (w.r.t. global $L_2$-norm). For Metattack with SGD instead of Adam for training the GCN, we use an SGD learning rate of 1 and restrict the training to $E_{\text{train}} = 100$ epochs.

## E.1  Jaccard-GCN

**Defense.** Additionally to a GCN, Jaccard-GCN [48] preprocesses the adjacency matrix. It computes the Jaccard coefficient of the binarized features for the pair of nodes of every edge, i.e., $\mathbf{J}_{ij} = \frac{\mathbf{X}_i\mathbf{X}_j}{\min\{\mathbf{X}_i+\mathbf{X}_j,1\}}$. Then edges are dropped where $\mathbf{J}_{ij} \leq \epsilon$.

**Adaptive attack.** We do not need to adapt gradient-based attacks as the gradient is equal to zero for dropped edges. Straightforwardly, we adapt Nettack to only consider non-dropped edges. Analogously, we ignore these edges in the Greedy Brute Force attack for increased efficiency.

## E.2  SVD-GCN

**Defense.** SVD-GCN [12] preprocesses the adjacency matrix with a low-rank approximation (LRA) for a fixed rank $r$, utilizing the Singular Value Decomposition (SVD) $\mathbf{A} = \mathbf{U}\boldsymbol{\Sigma}\mathbf{V}^{\top} \approx \mathbf{U}_r\boldsymbol{\Sigma}_r\mathbf{V}_r^{\top} = \mathbf{A}_r$. Note that the LRA is performed on $\mathbf{A}$ before adding self-loops and GCN-normalization (see above). Thereafter, the dense $\mathbf{A}_r$ is passed to the GCN as usual. Since $\mathbf{A}$ is symmetric and positive semi-definite, we interchangeably refer to the singular values/vectors also as eigenvalues/eigenvectors.

**Adaptive attack.** Unfortunately, the process of determining the singular vectors $\mathbf{U}_r$ and $\mathbf{V}_r$ is highly susceptible to small perturbations, and so is its gradient. Thus, we circumvent the need of differentiating the LRA.

We now explain the approach from a geometrical perspective. Each row of $\mathbf{A}$ (or interchangeably column as $\mathbf{A}$ is symmetric) is interpreted as coordinates of a high-dimensional point. The $r$ most significant eigenvectors of $\mathbf{A}$ span an $r$-dimensional subspace, onto which the points are projected by the LRA. Adding or removing an adversarial edge $(i,j)$ corresponds to moving the point $\mathbf{A}_i$ along dimension $j$, i.e., $\mathbf{A}_i \pm \mathbf{e}_j$ (vice-versa for $\mathbf{A}_j$). As hinted at in § 4, the $r$ most significant eigenvectors of $\mathbf{A}$ turn out to usually have few large components. Thus, the relevant subspace is mostly aligned with only few dimensions.

Changes along the highest-valued eigenvectors are consequently preserved by LRA. To quantify how much exactly such a movement along a dimension $j$, i.e., $\mathbf{e}_j$, is preserved, we project the movement itself onto the subspace and extract the projected vector's $j$-th component. More formally, we denote the projection matrix onto the subspace as $\mathbf{P} = \sum_{k=0}^{r} \mathbf{v}_k\mathbf{v}_k^T$ where $\mathbf{v}_k$ are the eigenvectors of $\mathbf{A}$. We now score each dimension $j$ with $(\mathbf{P}\mathbf{e}_j)_j = \mathbf{P}_{jj}$. Since the adjacency matrix is symmetric and rows and columns are hence exchangeable, we then symmetrize the scores $\mathbf{W}_{ij} = (\mathbf{P}_{ii} + \mathbf{P}_{jj})/2$.

Finally, we decompose the perturbed adjacency matrix $\tilde{\mathbf{A}} = \mathbf{A} + \delta\mathbf{A}$ and, thus, only need gradients for $\delta\mathbf{A}$. Using the approach sketched above, we now replace $\text{LRA}(\mathbf{A} + \delta\mathbf{A}) \approx \text{LRA}(\mathbf{A}) + \delta\mathbf{A} \circ \mathbf{W}$.

The weights $\mathbf{W}$ can also be incorporated into the Greedy Brute Force attack by dropping edges with weight $< 0.2$ and, for efficient early stopping, sort edges to try in order of descending weight. Similarly, Nettack's score function $s_{\text{struct}}(i,j)$ – which attains positive and negative values, while $\mathbf{W}$ is positive – can be wrapped to $s'_{\text{struct}}(i,j) = \log(\exp(s_{\text{struct}}(i,j)) \circ \mathbf{W}) = s_{\text{struct}}(i,j) + \log \mathbf{W}$.

Note that we assume that the direction of the eigenvectors remains roughly equal after perturbing the adjacency matrix. In practice, we find this assumption to be true. Intuitively, a change along the dominant eigenvectors should even reinforce their significance.

### E.3 RGCN

**Defense.** The implementations of R(obust)GCN provided by the authors[5] and in the widespread DeepRobust [35] library[6] are both consistent, but diverge slightly from the paper [63]. We use and now present RGCN according to those reference implementations. Principally, RGCN models the hidden states as Gaussian vectors with diagonal variance instead of sharp vectors. In addition to GCN's $\mathbf{A}'$, a second $\mathbf{A}'' = (\mathbf{D} + \mathbf{I})^{-1}(\mathbf{A} + \mathbf{I})(\mathbf{D} + \mathbf{I})^{-1}$ is prepared to propagate the variances. The mean and variance of this hidden Gaussian distribution are initialized as $\mathbf{M}^{(0)} = \mathbf{V}^{(0)} = \mathbf{X}$. Each layer first computes an intermediate distributions given by $\hat{\mathbf{M}}^{(l)} = \text{elu}(\text{dropout}(\mathbf{M}^{(l-1)})\mathbf{W}_M^{(l-1)})$ and $\hat{\mathbf{V}}^{(l)} = \text{relu}(\text{dropout}(\mathbf{V}^{(l-1)})\mathbf{W}_V^{(l-1)})$. Then, attention coefficients $\boldsymbol{\alpha}^{(l)} = e^{-\gamma \hat{\mathbf{V}}^{(l)}}$ are calculated with the aim to subdue high-variance dimensions (where exponentiation is element-wise and $\gamma$ is a hyperparameter). The final distributions are obtained with $\mathbf{M}^{(l)} = \mathbf{A}'\hat{\mathbf{M}}'^{(l)} \circ \boldsymbol{\alpha}^{(l)}$. Note the absence of bias terms. After the last layer, point estimates are sampled from the distributions via the reparameterization trick, i.e., scalars are sampled from a standard Gaussian and arranged in a matrix $\mathbf{R}$. These samples are then used to obtain the logits via $\mathbf{M}^{(L)} + \mathbf{R} \circ (\mathbf{V}^{(L)} + \epsilon)^{\frac{1}{2}}$ (where the square root applies element-wise and $\epsilon$ is a hyperparameter). Adam is the default optimizer. The loss is extended with the regularizer $\beta \sum_i \text{KL}(\mathcal{N}(\hat{\mathbf{M}}_i^{(1)}, \text{diag}(\hat{\mathbf{V}}_i^{(1)})) \| \mathcal{N}(\mathbf{0}, \mathbf{I}))$ (where $\beta$ is a hyperparameter).

**Adaptive attack.** A direct gradient attack suffices for a strong adaptive attack. Only when unrolling the training procedure for Metattack and Meta-PGD, we increase hyperparameter $\epsilon$ from $10^{-8}$ to $10^{-2}$ to retain numerical stability.

### E.4 ProGNN

**Defense.** We use and present Pro(perty)GNN [30] exactly following the implementation provided by the authors in their DeepRobust [35] library[6]. ProGNN learns an alternative adjacency matrix $\mathbf{S}$ that is initialized with $\mathbf{A}$. A regular GCN – which, as usual, adds self-loops and applies GCN-normalization – is trained using $\mathbf{S}$, which is simultaneously updated in every $\tau$-th epoch. For that, first a gradient descent step is performed on $\mathbf{S}$ with learning rate $\eta$ and momentum $\mu$ towards minimizing the principal training loss alongside two regularizers that measure deviation $\beta_1 \|\mathbf{S} - \mathbf{A}\|_F^2$ and feature smoothness $\frac{\beta_2}{2} \sum_{i,j} \mathbf{S}_{ij} \| \frac{\mathbf{X}_i}{\sqrt{d_i}} - \frac{\mathbf{X}_j}{\sqrt{d_j}} \|^2$ (where $d_i = \sum_j \mathbf{S}_{ij} + 10^{-3}$). Next, the singular value decomposition $\mathbf{U}\boldsymbol{\Sigma}\mathbf{V}^T$ of the updated $\mathbf{S}$ is computed, and $\mathbf{S}$ is again updated to be $\mathbf{U}\max(0, \boldsymbol{\Sigma} - \eta\beta_3)\mathbf{V}^T$ to promote low-rankness. Thereafter, $\mathbf{S}$ is again updated to be $\text{sgn}(\mathbf{S}) \circ \max(0, |\mathbf{S}| - \eta\beta_4)$ to promote sparsity. Finally, the epoch's resulting $\mathbf{S}$ is obtained by clamping its elements between $0$ and $1$.

**Adaptive attack.** Designing an adaptive attack for ProGNN proved to be a challenging endeavor. We describe the collection of tricks in § 4's Example 2.

### E.5 GNNGuard

**Defense.** We closely follow the authors' implementation[7] as it deviates from the formal definitions in the paper [58]. GNNGuard adopts a regular GCN and, before each layer, it adaptively weights down alleged adversarial edges. Thus, each layer has a unique propagation matrix $\mathbf{A}^{(l)}$ that is used instead of $\mathbf{A}'$.

---

[5] https://github.com/ZW-ZHANG/RobustGCN     [6] https://github.com/DSE-MSU/DeepRobust
[7] https://github.com/mims-harvard/GNNGuard

GNNGuard's rule-based edge reweighting can be clustered into four consecutive steps: (1) the edges are reweighted based on the pair-wise cosine similarity $\mathbf{C}_{ij}^{(l)} = \frac{\mathbf{H}_i^{(l-1)} \cdot \mathbf{H}_j^{(l-1)}}{\|\mathbf{H}_i^{(l-1)}\| \|\mathbf{H}_j^{(l-1)}\|}$ according to $\mathbf{S}^{(l)} = \mathbf{A} \circ \mathbf{C}^{(l)} \circ \mathbb{I}[\mathbf{C}^{(l)} \geq 0.1]$, where edges with too dissimilar node embeddings are removed (see Iverson bracket $\mathbb{I}[\mathbf{C}^{(l)} \geq 0.1]$). Then, (2) the matrix is rescaled $\mathbf{\Gamma}_{ij}^{(l)} = \mathbf{s}_{ij}^{(l)}/\mathbf{s}_i^{(l)}$ with $\mathbf{s}_i^{(l)} = \sum_j \mathbf{S}_{ij}^{(l)}$ For stability, if $\mathbf{s}_i^{(l)} < \epsilon$, $\mathbf{s}_i^{(l)}$ is set to 1 (here $\epsilon$ is a small constant). Next, (3) self-loops are added and $\mathbf{\Gamma}^{(l)}$ is non-linarily transformed according to $\hat{\mathbf{\Gamma}}^{(l)} = \exp_{\neq 0}(\mathbf{\Gamma}^{(l)} + \operatorname{diag} 1/1 + \mathbf{d}^{(l)})$, where $\exp_{\neq 0}$ only operates on nonzero elements and $\mathbf{d}_i^{(l)} = \|\mathbf{\Gamma}_i^{(l)}\|_0$ is the row-wise number of nonzero entries. Last, (4) the result is smoothed over the layers with $\mathbf{\Omega}^{(l)} = \sigma(\rho)\mathbf{\Omega}^{(l-1)} + (1 - \sigma(\rho))\hat{\mathbf{\Gamma}}^{(l)}$ with learnable parameter $\rho$ and sigmoid function $\sigma(\cdot)$.

The resulting reweighted adjacency matrix $\mathbf{\Omega}^{(l)}$ is then GCN-normalized (without adding self-loops) and passed on to a GCN layer. Note that steps (1) to (3) are excluded from back-propagation during training. When comparing with the GNNGuard paper, one notices that among other deviations, we have omitted learnable edge pruning because it is disabled in the reference implementation.

**Adaptive attack.** The hyperparameter $\epsilon$ must be increased from $10^{-6}$ to $10^{-2}$ during the attack to retain numerical stability. In contrast to the reference implementation but as stated above, it is important to place the hard filtering step $\mathbb{I}[\mathbf{C}^{(l)} \geq 0.1]$ for $\mathbf{S}^{(l)}$ s.t. the gradient calculation w.r.t. $\mathbf{A}$ is not suppressed for these entries.

### E.6 GRAND

**Defense.** The Graph Random Neural Network (GRAND) [15] model is the only defense from our selection that is not based on a GCN. First, $\mathbf{A}$ is endowed with self-loops and GCN-normalized to obtain $\mathbf{A}'$. Also, each row of $\mathbf{X}$ is $l_1$-normalized, yielding $\mathbf{X}'$. Next, rows from $\mathbf{X}'$ are randomly dropped with probability $\delta$ during training to generate a random augmentation, and $\mathbf{X}'$ is scaled by $1 - \delta$ during inference to compensate, thereby obtaining $\hat{\mathbf{X}}$. Those preprocessed node features are then propagated multiple times along the graph to get $\overline{\mathbf{X}} = \frac{1}{K+1} \sum_{k=0}^{K} \mathbf{A}'^k \hat{\mathbf{X}}$. Finally, dropout is applied once to $\overline{\mathbf{X}}$, and the result is plugged into a 2-layer MLP with dropout and ReLU activation to obtain class probabilities $\mathbf{Z}$. The authors also propose an alternative architecture using a GCN instead of an MLP, however, we do not explore this option since the MLP version is superior according to their own results.

GRAND is trained with Adam. The training loss comprises the mean of the cross-entropy losses of $S$ model evaluations, thereby incorporating multiple random augmentations. Additionally, a consistency regularizer is added to enforce similar class probabilities across all evaluations. More formally, first the probabilities are averaged across all evaluations: $\overline{\mathbf{Z}} = \frac{1}{S} \sum_{s=1}^{S} \mathbf{Z}^{(s)}$. Next, each node's categorical distribution is sharpened according to a temperature hyperparameter $T$, i.e., $\overline{\mathbf{Z}}'_{ij} = \overline{\mathbf{z}}_{ij}^{\frac{1}{T}} / \sum_c \overline{\mathbf{z}}_{ic}^{\frac{1}{T}}$. The final regularizer penalizes the distance between the class probabilities and the sharpened averaged distributions, namely $\frac{\beta}{S} \sum_{s=1}^{S} \|\mathbf{Z}^{(s)} - \overline{\mathbf{Z}}'\|_F^2$.

**Adaptive attack.** When unrolling the training procedure for Metattack and Meta-PGD, to reduce the memory footprint, we reduce the number of random augmentations per epoch to 1, and we use a manual gradient calculation for the propagation operation. We also initialize Meta-PGD with a strong perturbation found by Meta-PGD on ProGNN. Otherwise, the attack has issues finding a perturbation with high loss; it presumably stalls in a local optimum. It is surprising that "only" initializing from GCN instead of ProGNN does not give a satisfyingly strong attack. Finally, we use the same random seed for every iteration of Metattack and Meta-PGD, as otherwise the constantly changing random graph augmentations make the optimization very noisy.

### E.7 Soft-Median-GDC

**Defense.** The Soft-Median-GDC [17] deviates in two ways from a GCN: (1) it uses Personalized Page Rank (PPR) with restart probability $\alpha = 0.15$ to further preprocess the adjacency matrix after adding self-loops and applying GCN-normalization. The result is then sparsified using a row-wise top-$k$ operation ($k = 64$). (2) the message passing aggregation is replaced with a robust estimator

called Soft-Median. From the perspective of node $i$, a GCN uses the message passing aggregation $\mathbf{H}_i^{(l)} = \mathbf{A}_i\mathbf{H}^{(l-1)}$ which can be interpreted as a weighted mean/sum. In Soft-Median-GDC, the "weights" $\mathbf{A}_i$ are replaced with a scaled version of $\mathbf{A}_i \circ \mathrm{softmax}\left(-\mathbf{c}/T\sqrt{d}\right)$. Here the vector $\mathbf{c}$ denotes the distance between hidden embedding of a neighboring node to the neighborhood-specific weighted dimension-wise median: $\mathbf{c}_i = \|\mathrm{Median}(\mathbf{A}_i, \mathbf{H}^{(l-1)}) - \mathbf{H}_i^{(l-1)}\|$. To keep the scale, these weights are scaled s.t. they sum up to $\sum \mathbf{A}_i$.

**Adaptive attack.** During gradient-based attacks, we adjust the $\mathbf{c}$ of every node s.t. it now captures the distance to all other nodes, not only neighbors. This of course modifies the values of $\mathbf{c}$, but is necessary to obtain a nonzero gradient w.r.t. to all candidate edges. We initialize PGD with a strong perturbation found by a similar attack on GCN, and initialize Meta-PGD with a perturbation from a similar attack on ProGNN (as with GRAND, using an attack against GCN as a base would be insufficient here).

## F    Evaluation of adaptive attacks

In Table F.1, we summarize the variants of the datasets we use, both of which we have precisely extracted from Nettack's code[8]. In Fig. F.1, we complement Fig. 2 and compare the (R)AUC of all defenses on Citeseer. The robustness estimates for the defenses on Citeseer are also much lower as originally reported. For completeness, we give absolute envelope curve plots for all settings and datasets as well as for higher budgets in Fig. F.2 and Fig. F.3 (compare with Fig. 4 and Fig. 5).

Table F.1: Statistics of the datasets we used. We measure homophily as the fraction of edges which connect nodes of the same class.

| Dataset | Nodes | Undirected Edges | Features | Classes | Avg. Degree | Homophily |
|---|---|---|---|---|---|---|
| Cora ML [2] | 2485 | 5069 | 1433 | 7 | 4.08 | 0.804 |
| Citeseer [19] | 2110 | 3668 | 3703 | 6 | 3.477 | 0.736 |

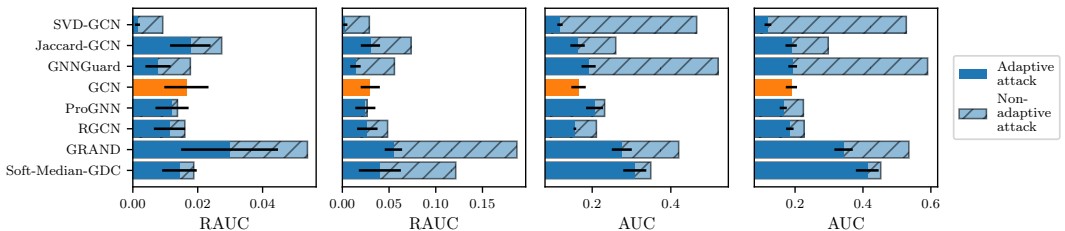

(a) Global, Poisoning   (b) Global, Evasion   (c) Local, Poisoning   (d) Local, Evasion

Figure F.1: Variant of Fig. 2 for Citeseer.

[8]  https://github.com/danielzuegner/nettack

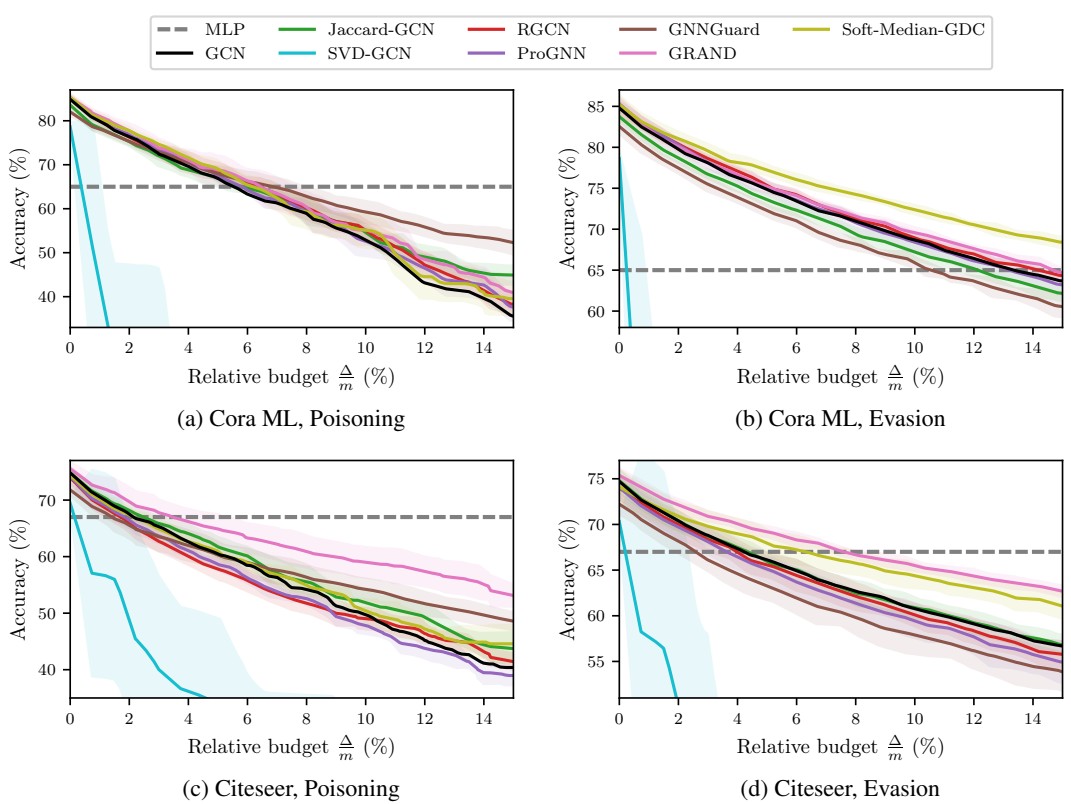

(a) Cora ML, Poisoning

(b) Cora ML, Evasion

(c) Citeseer, Poisoning

(d) Citeseer, Evasion

Figure F.2: Absolute variant of Fig. 4, showing relative budgets up to 15%.

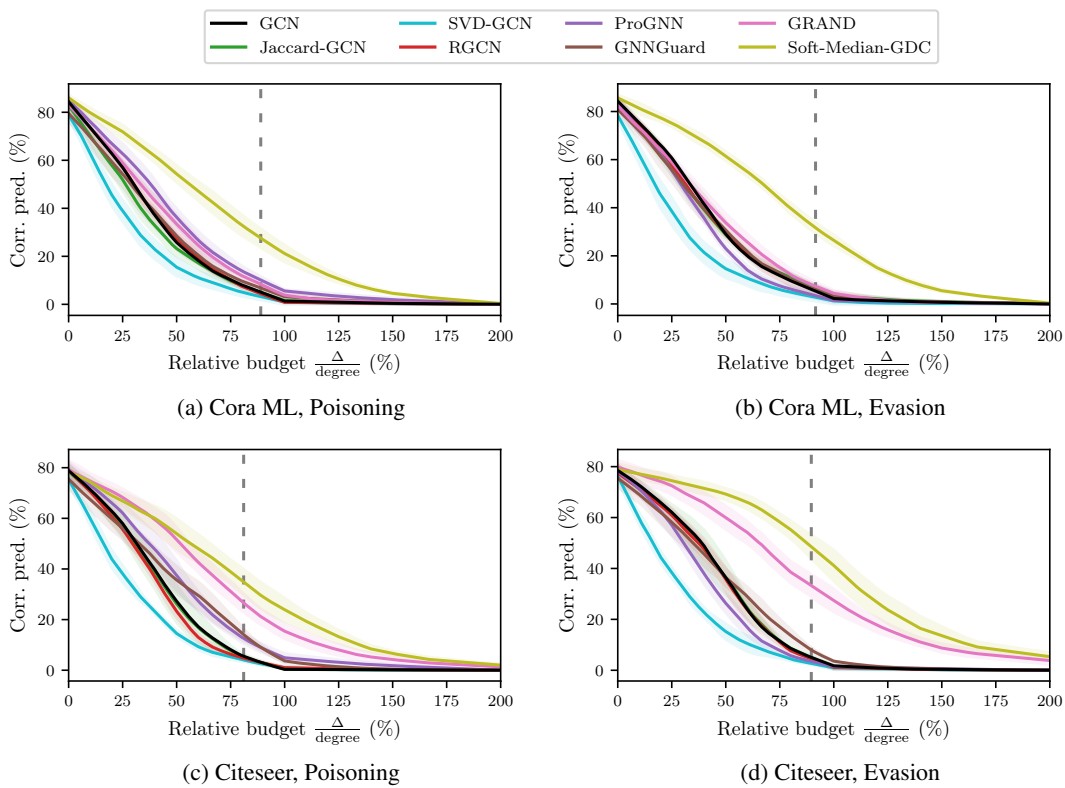

(a) Cora ML, Poisoning

(b) Cora ML, Evasion

(c) Citeseer, Poisoning

(d) Citeseer, Evasion

Figure F.3: Absolute variant of Fig. 5, showing relative budgets up to 200%.

# G Ensemble transferability study

In Fig. 8, we transfer attacks found on an *individual* model to other models. It is natural to also assess the strength of transfer attacks supplied by *ensembles* of models. In Fig. G.1, we address this question for 2-ensembles. For poisoning, the combination of RGCN and ProGNN turns out to be (nearly) the strongest in all cases, which is reasonable since both already form strong individual transfer attacks as is evident in Fig. 8. For evasion, the differences are more subtle.

We also investigate 3-ensembles, but omit the plots due to their size. For poisoning, RGCN and ProGNN now combined with Soft-Median-GDC remain the strongest transfer source, yet the improvement over the 2-ensemble is marginal. For evasion, there is still no clear winner.

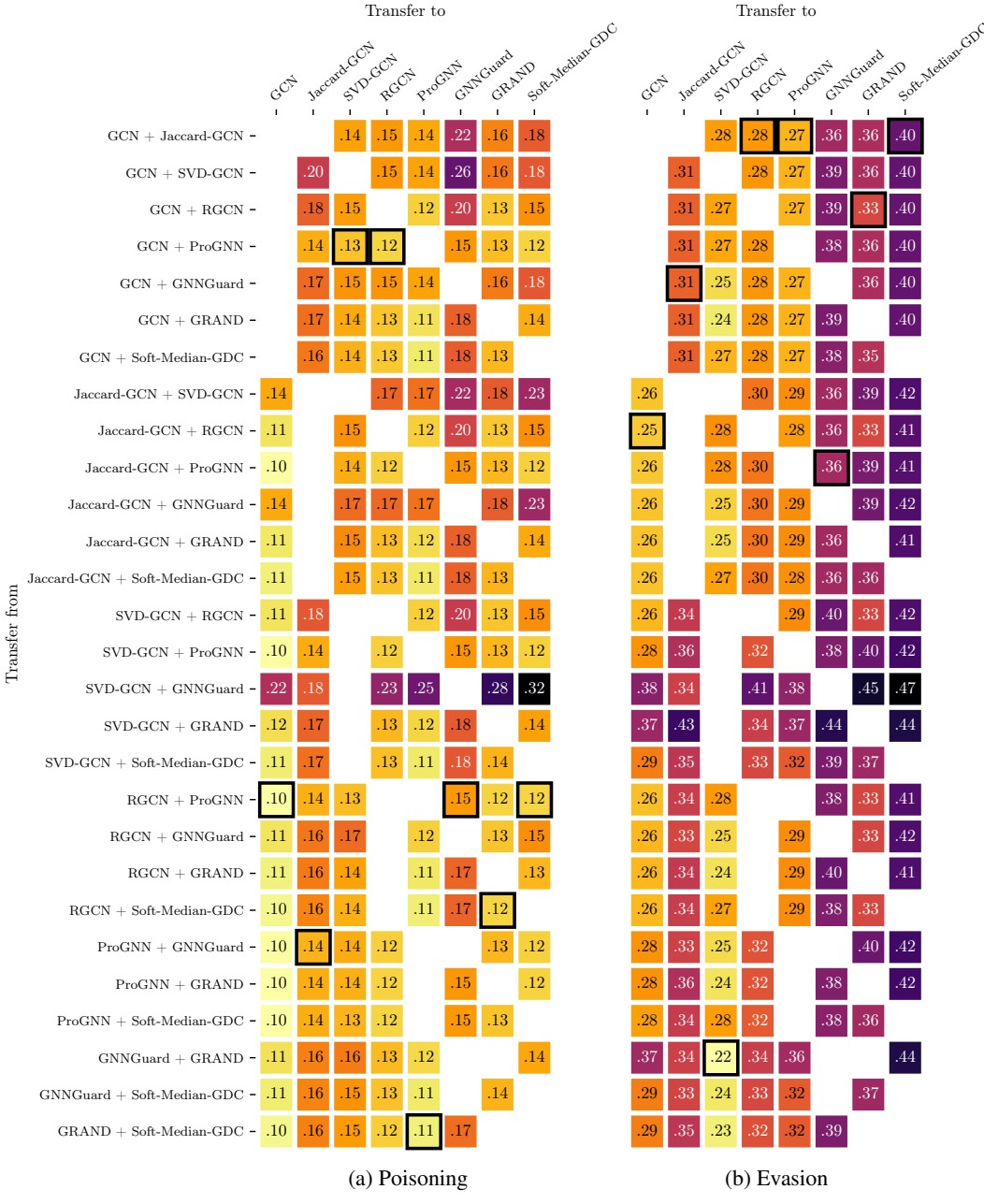

Figure G.1: Variant of Fig. 8 with ensembles of models as attack transfer sources. The color maps are *not* matched across (a) and (b) for improved readability.

# H GCN and defense hyperparameters: original vs. tuned for adaptive attacks

To allow for the fairest comparison possible, we tuned the hyperparameters for each model (including GCN) towards maximizing both clean accuracy and adversarial robustness on a single random data split. In Table H.1, we list all hyperparameter configurations. While we cannot run an exhaustive search over all hyperparameter settings, we report substantial gains for most defenses and the GCN in Fig. H.1. The only exceptions are GRAND, Soft-Median-GDC on Cora ML, and GNNGuard. For GRAND, we do not report results for the default hyperparameters as they did not yield satisfactory clean accuracy. Moreover, for Soft-Median-GDC on Cora ML and GNNGuard we were not able to substantially improve over the default hyperparameters.

For the GCN, tuning is important to ensure that we have a fair and equally-well tuned baseline. A GCN is the natural baseline since most defense methods propose slight modifications of a GCN or additional steps to improve the robustness. For the defenses, tuning is vital since they were most originally tuned w.r.t. non-adaptive attacks. In any case, the tuning should counterbalance slight variations in the setup.

As stated in the introduction, each attack only provides an upper bound for the actual adversarial robustness of a model (with fixed hyperparameters). A future attack of increased efficacy might lead to a tighter estimate. Thus, when we empirically compare the defenses to a GCN, we only compare upper bounds of the respective actual robustness. However, we attack the GCN with state-of-the-art approaches that were developed by multiple researchers specifically for a GCN. Even though we also tune the parameters of the adaptive attacks, we argue that the robustness estimate for a GCN is likely tighter than our robustness estimate for the defenses. In summary, the tuning of hyperparameters is necessary that we can fairly compare the robustness of multiple models, even though, we only compare upper bounds of the true robustness.

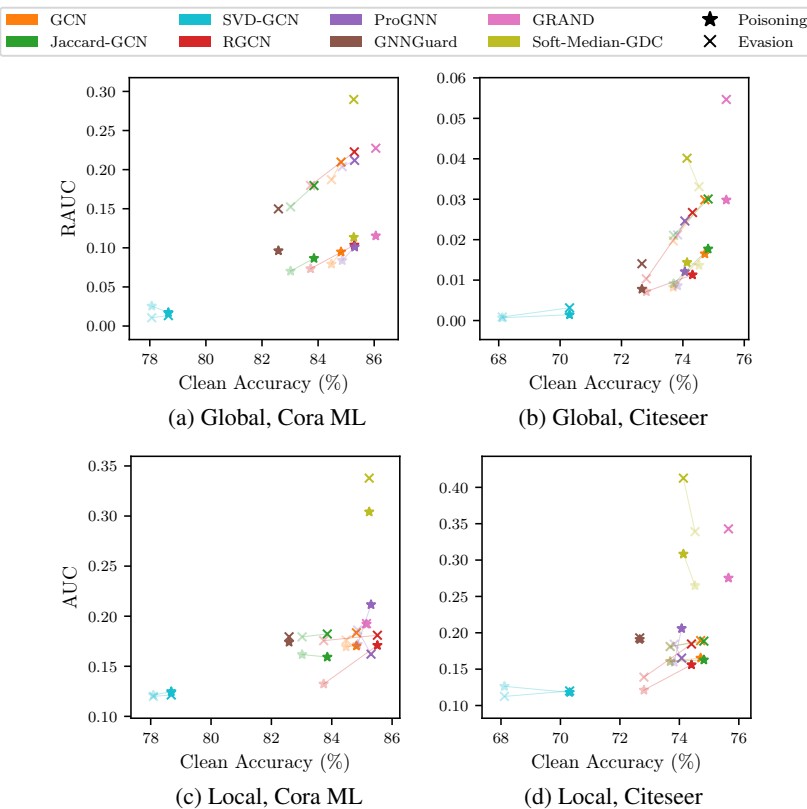

Figure H.1: Each defense's clean accuracy vs. (R)AUC values of the strongest attacks, akin to Fig. 6. Muted (semi-transparent) colors represent untuned defenses (except for Soft-Median-GDC on Cora ML and GNNGuard), solid colors denote tuned defenses, and lines connect the two. Our tuned defenses are almost always better than untuned variants w.r.t. both clean accuracy and robustness.

Table H.1: GCN and defense hyperparameters.

**GCN**

| Tuned | Hidden | Dropout | Max epochs | Patience | LR | $L_2$ reg. |
|---|---|---|---|---|---|---|
| × | $1 \times 16$ | 0.5 | 3000 | 50 | 0.01 | 0.0005 |
| ✓ | $1 \times 64$ | 0.9 | 3000 | 50 | 0.01 | 0.001 |

**Jaccard-GCN**

| Tuned | Hidden | Dropout | $\epsilon$ | Max epochs | Patience | LR | $L_2$ reg. |
|---|---|---|---|---|---|---|---|
| × | $1 \times 16$ | 0.5 | 0.0 | 3000 | 200 | 0.01 | 0.0005 |
| ✓ | $1 \times 64$ | 0.9 | 0.0 | 3000 | 50 | 0.01 | 0.001 |

**SVD-GCN**

| Tuned | Hidden | Dropout | Rank | Max epochs | Patience | LR | $L_2$ reg. |
|---|---|---|---|---|---|---|---|
| × | $1 \times 16$ | 0.5 | 50 | 3000 | 200 | 0.01 | 0.0005 |
| ✓ | $1 \times 64$ | 0.9 | 50 | 3000 | 50 | 0.01 | 0.001 |

**RGCN**

| Tuned | Hidden | Dropout | $\epsilon$ | $\gamma$ | Max epochs | Patience | LR | $L_2$ reg. | $\beta$ |
|---|---|---|---|---|---|---|---|---|---|
| × | $1 \times 16$ | 0.6 | 1e-8 | 1.0 | 3000 | 50 | 0.01 | 0.0005 | 0.0005 |
| ✓ | $1 \times 32$ | 0.6 | 1e-8 | 1.0 | 3000 | 50 | 0.01 | 0.01 | 0.0005 |

**ProGNN**

| Tuned | | Hidden | Dropout | Max epochs | Patience | LR | $L_2$ reg. | $\tau$ | $\eta$ | $\mu$ | $\beta_1$ | $\beta_2$ | $\beta_3$ | $\beta_4$ |
|---|---|---|---|---|---|---|---|---|---|---|---|---|---|---|
| × | Cora ML | $1 \times 16$ | 0.5 | 3000 | 50 | 0.01 | 0.0005 | 2 | 0.01 | 0.9 | 1.0 | 0.001 | 1.5 | 0.0005 |
| × | Citeseer | $1 \times 16$ | 0.5 | 3000 | 50 | 0.01 | 0.0005 | 2 | 0.01 | 0.9 | 1.0 | 0.0001 | 1.5 | 0.0005 |
| ✓ | Cora ML | $1 \times 16$ | 0.5 | 3000 | 50 | 0.01 | 0.0005 | 2 | 0.01 | 0.9 | 1.0 | 0.1 | 10.0 | 0.1 |
| ✓ | Citeseer | $1 \times 16$ | 0.5 | 3000 | 50 | 0.01 | 0.0005 | 2 | 0.01 | 0.9 | 1.0 | 0.2 | 20.0 | 0.2 |

**GNNGuard**

| Tuned | Hidden | Dropout | Pruning | $\epsilon$ | Max epochs | Patience | LR | $L_2$ reg. |
|---|---|---|---|---|---|---|---|---|
| × | $1 \times 16$ | 0.5 | × | 1e-6 | 81 | n/a | 0.01 | 0.0005 |

**GRAND**

| Tuned | | Hidden | Dropout | $\overline{\mathbf{X}}$ dropout | $\delta$ | $K$ | Max epochs | Patience | LR | $L_2$ reg. | $S$ | $\beta$ | $T$ |
|---|---|---|---|---|---|---|---|---|---|---|---|---|---|
| ✓ | Cora ML | $1 \times 32$ | 0.5 | 0.5 | 0.5 | 8 | 3000 | 50 | 0.05 | 0.0001 | 4 | 1.0 | 0.5 |
| ✓ | Citeseer | $1 \times 32$ | 0.2 | 0.0 | 0.5 | 2 | 3000 | 50 | 0.05 | 0.0005 | 2 | 0.7 | 0.3 |

**Soft-Median-GDC**

| Tuned | | Hidden | Dropout | $k$ | $\alpha$ | $T$ | Max epochs | Patience | LR | $L_2$ reg. |
|---|---|---|---|---|---|---|---|---|---|---|
| × | | $1 \times 64$ | 0.5 | 64 | 0.15 | 0.5 | 3000 | 50 | 0.01 | 0.001 |
| ✓ | Citeseer | $1 \times 64$ | 0.5 | 64 | 0.25 | 0.5 | 3000 | 50 | 0.01 | 0.001 |

# I  Comparison of success of attack approaches

In Fig. I.1 we report which of the global attack techniques generate the strongest attacks, and in Fig. I.3, we break down every global attack attempt. Analogously, in Fig. I.2 and Fig. I.4, we report which local attack techniques require the smallest budget to misclassify the target nodes. In Fig. I.3, we additionally compare different loss types for global attacks.

In general, we can say that PGD is the dominating attack for global evasion. For poisoning, Meta-PGD seems to be the strongest – slightly more successful than Metattack, though not in every case. Greedy brute force dominates the local attacks, but for some defenses, PGD and Nettack have an edge.

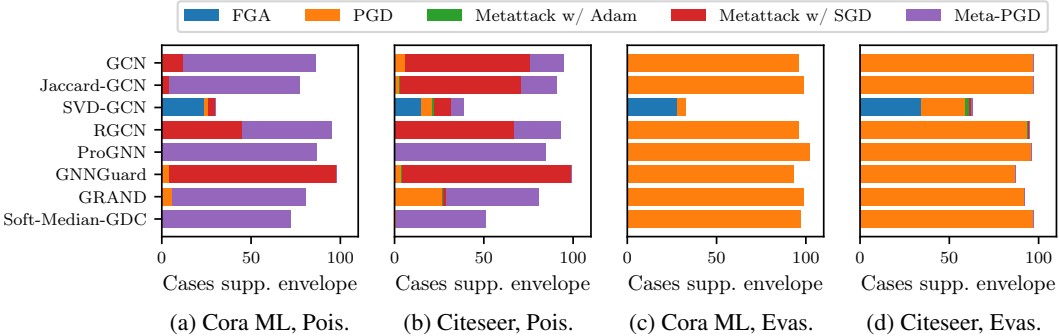

Figure I.1: Number of global attack attempts which support the envelope curve over all attack attempts, as introduced in Fig. 3. We observe that for evasion, PGD almost always yields the strongest attack, while for poisoning, either Metattack, Meta-PGD, or both dominate. Using Adam instead of SGD to train the defense nearly always worsens Metattack's performance.

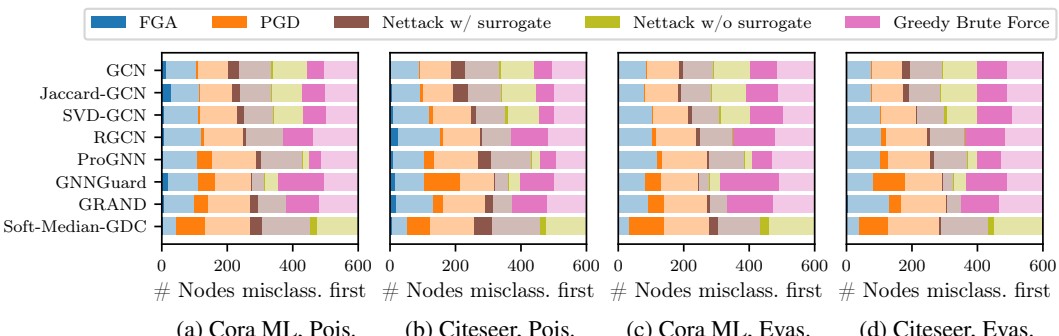

Figure I.2: Number of target nodes for which the respective local attack needs the least budget (among all attacks) to misclassify them. When multiple attacks achieve the same lowest budget, the target node is counted in parts towards each winning attack and drawn with a muted color. We observe that greedy brute force is often the strongest attack; only sometimes, PGD and Nettack beat it on some defenses, especially for poisoning. Using the defense's weights instead of a surrogate model for Nettack is rarely an improvement. Still, for the majority of target nodes, multiple attacks are equally strong in terms of achieving the same lowest budget (tie). We do not run the greedy brute force attack on Soft-Median-GDC due to the costly PPR calculation.

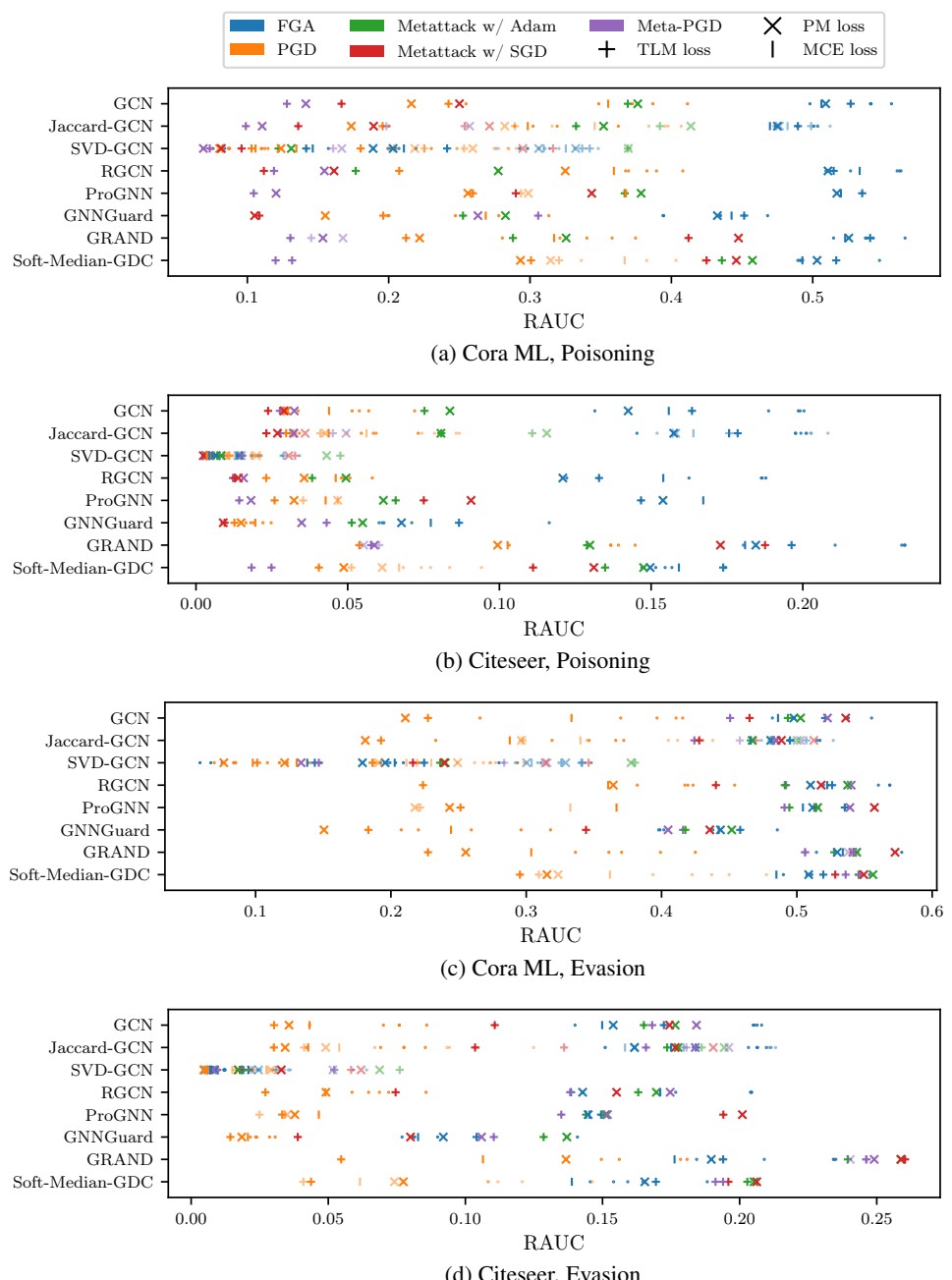

Figure I.3: The RAUC of every global attack we have conducted. Attacks are color-coded by principal technique, and markers indicate the attack loss. Muted colors represent attacks *without* edge masking (Jaccard-GCN), our edge weighting trick (SVD-GCN), multiple PGD auxiliary models (ProGNN), Meta-PGD initialization from ProGNN and unlimited unrolled epochs (GRAND), and PGD initialization from GCN (Soft-Median-GDC). We observe that (1) the TLM and PM losses are superior in almost all cases; (2) PGD attacks are best for evasion while Metattack and Meta-PGD are unsuited; (3) Metattack with SGD and Meta-PGD are best for poisoning while Metattack w/ Adam even falls behind the surprisingly strong evasion-poisoning transfer; (4) FGA is weak for each defense apart from SVD-GCN; (5) the cited adaptions are beneficial as attacks with muted colors are worse; (6) a strong adaptive attack is necessary to reach a low RAUC.

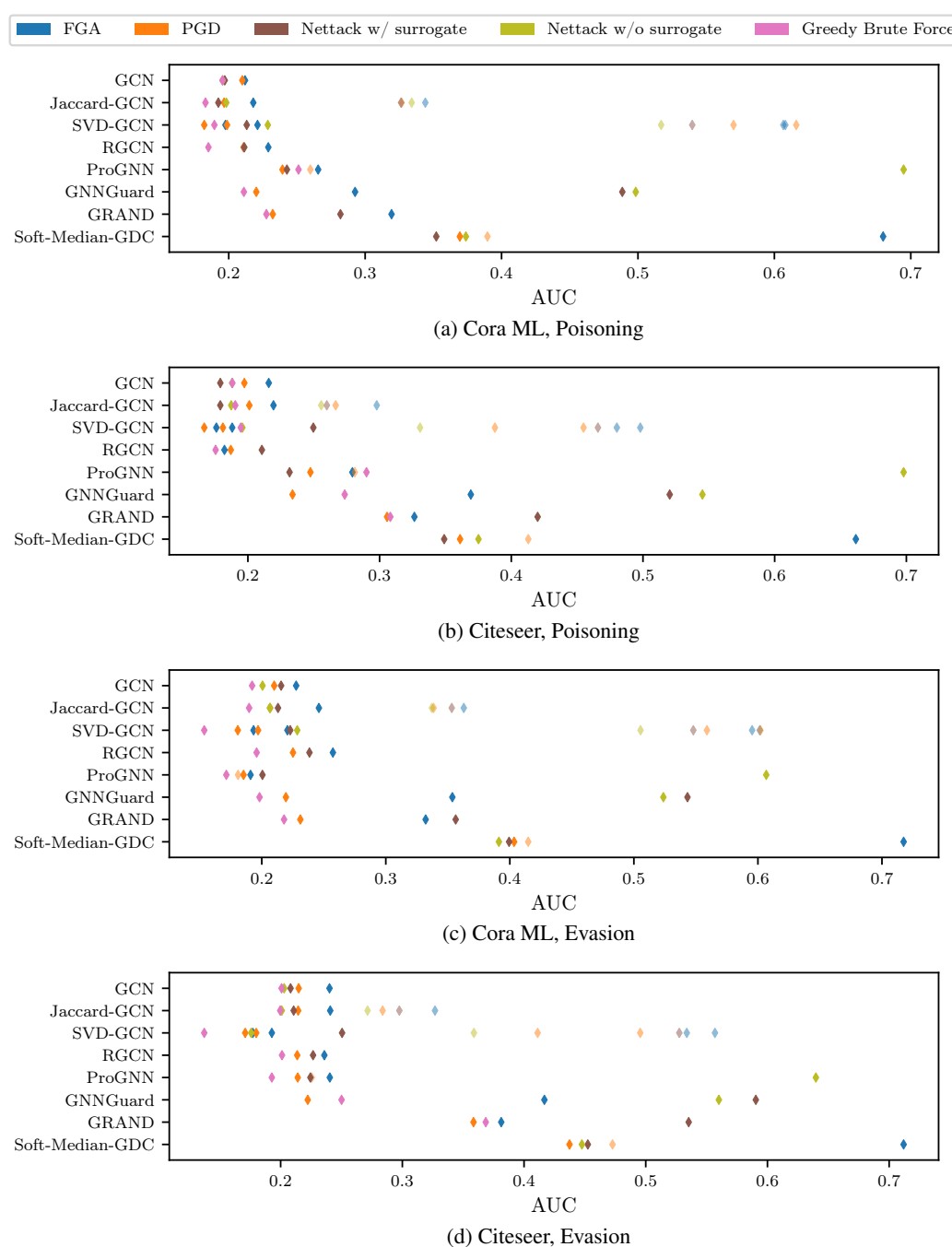

Figure I.4: The AUC of every local attack we have conducted. Attacks are color-coded by principal technique. Muted colors have the same signification as in Fig. I.3. We observe that (1) greedy brute force is often the best attack, closely followed by PGD, while FGA is not as strong; (2) Nettack can rarely be made stronger by utilizing the target model's weights instead of a surrogate model (red); (3) many defenses successfully defend against Nettack; (4) against those defenses for which we have adapted Nettack, it becomes much stronger (muted vs. normal green); (5) the adaptions are also beneficial for other attacks, as those with muted colors are worse.

## J    Sensitivity to random seed

When transferring perturbations from evasion to poisoning, a different random seed is used for training the poisoned model than was used for the evasion one. In Fig. J.1, we study using the example of GCN and ProGNN whether poisoning success improves when we instead assume the same seed is used. This is indeed the case and turns out particularly strong on tuned ProGNN. However, by using multiple auxiliary models during evasion as detailed in § 4 under the ProGNN example subheading, we can substantially reduce the dependence of the attack upon a particular random seed and thereby improve attack performance.

## K    Robustness over node degree

We explore the behavior of nodes under attack depending on their degree. In Fig. K.1, we show the probability that a successfully misclassified node falls into a certain degree range, broken down by relativ budget.

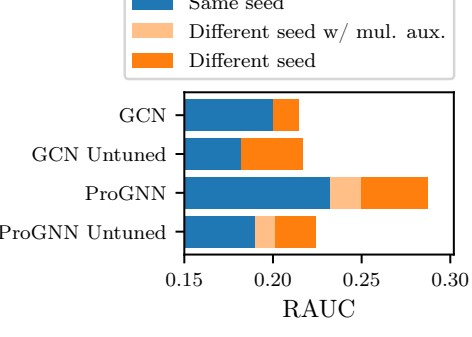

Figure J.1: Lowest RAUC achieved by global evasion-poisoning transfer attacks on Cora ML under the premise that the random seed used by the victim is known respectively unknown to the attacker. While not knowing the seed is disadvantageous especially on ProGNN, our attack using multiple auxiliary models successfully compensates this issue.

We cannot confirm the prevalent conjecture that global attacks tend to target low-degree nodes, as they are easier to break. Our results show that all degree groups are misclassified uniformly over all budgets. There is no clear preference for lower-degree nodes.

For local attacks, on the other hand, we indeed observe that the success rate of changing the predicted class is independent of the node degree if and only if using a relative budget. For example, when allowing a certain relative budget, e.g., 100% of the target node's degree, we manage to misclassify the same fraction of 1-degree target nodes (with absolute budget of 1) as 5-degree ones (with absolute budget of 5).

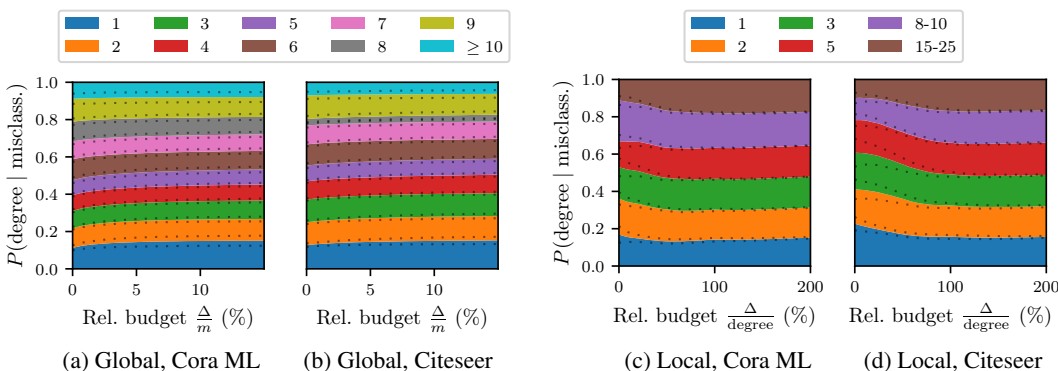

|  (a) Global, Cora ML  |  (b) Global, Citeseer  |  (c) Local, Cora ML  |  (d) Local, Citeseer  |

Figure K.1: The probability that a misclassified node is in a certain degree range. More specifically, for global attacks, that is which ratios of test set nodes from subsets with degree $1, 2, 3, \ldots, 9, \geq 10$ are misclassified per budget, normalized s.t. the stacked results sum to 1 everywhere. For local attacks, we show the amount of nodes from each target node set misclassified per budget, again normalized s.t. the stack sums to 1. Results are averaged over all experiments conducted (including evasion and poisoning) on tuned models. The dotted lines indicate standard deviation. We observe no substantial systematic bias towards the misclassification of low-degree nodes.

# L    Attack characteristics

Next, we present interesting patterns of the adversarial perturbations for each model/defense. We show the *(1) node degree*, *(2) closeness centrality*, *(3) homophily*, *(4) Jaccard similarity* of node attributes, and *(5) the ratio of removed edges* over the strongest edge perturbations in Fig. L.1. For statistics 1-4, we consider the pairs of nodes that were affected by an adversarial edge flip (i.e., insertion or removal). Here we average over the strongest attack found for each budget (without transferring attacks between defenses). Thus, the values indicate what characteristics are important for strong, adaptive attacks.

**(1) Node degree.** For global attacks, the degree tends to be lower than the average degree of the dataset as given in Table F.1. The higher average degree for local attacks might be influenced by the node selection. Interestingly, on SVD-GCN attacks connect very high-degree nodes, most likely because high-degree nodes correspond to dimensions represented by the most significant eigenvectors of $\mathbf{A}$ (see § 4 Example 1 and § E.2). The attacks exploit the sensitivity of SVD-GCN to perturbations of high-degree nodes. This could hint towards how adaptive attacks catastrophically break SVD-GCN.

**(2) Closeness centrality.** The closeness centrality of a particular node $v$ is one over the sum of distances from $v$ to all other nodes in the graph, multiplied by the total number of nodes in the graph. Attacks against SVD-GCN connect very central nodes, which probably correlates with them having high degrees. Interestingly, also the perturbations for GNN-Guard seem to be of slightly increased centrality.

**(3) Homophily** refers here to the fraction of pairs of nodes that share the same class. Successful adaptive attacks on Jaccard-GCN share the same homophily as those on GCN, indicating that the Jaccard coefficient is not suited to filter heterophil edges. Attacks on SVD-GCN, GNNGuard, and Soft-Median-GDC have higher homophily than those on GCN, hinting that these defenses successfully filter some heterogeneous edges, forcing some attacks to adapt.

**(4) Jaccard similarity.** As expected, attacks on Jaccard-GCN have to compensate its filter

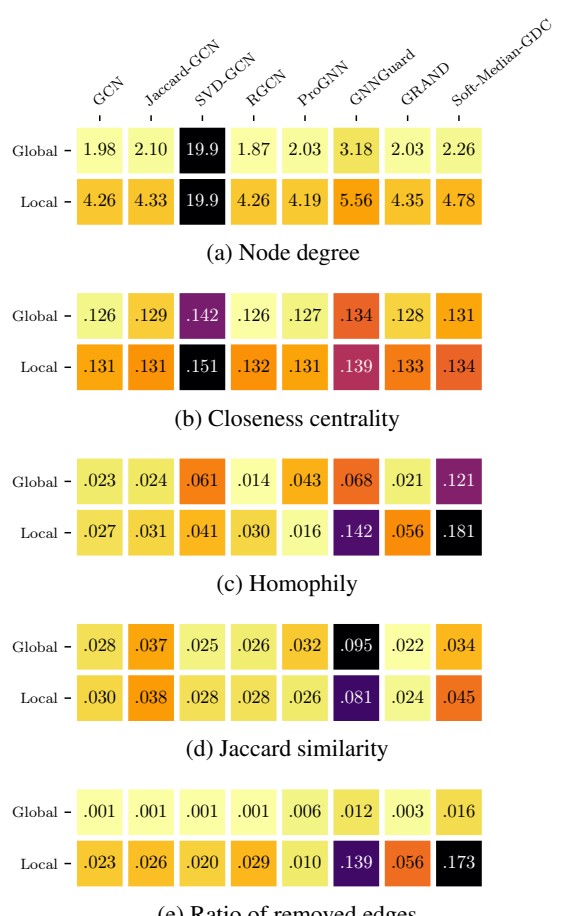

Figure L.1: Various metrics characterizing the nature of the adversarial edges from our strongest attacks, which are those visible in Fig. I.1 and Fig. I.2, as well as the nature of the nodes connected respectively disconnected by them.

by picking edges with nonzero coefficient. Attacks against GNNGuard connect nodes with very similar features, presumably to get past its cosine distance-based edge weighting. Curiously, attacks against Soft-Median-GDC behave similarly, yet only in the local setting and less pronounced. This is probably necessary to avoid that the new edges are weighted down as outliers by the robust aggregation, which becomes less of an issue when perturbing a large amount of edges in the global setting and thereby shifting what it means to be an outlier. Other defenses and especially GRAND admit connecting nodes as or more dissimilar than is the case on GCN.

**(5) Ratio of removed edges.** It is clear to see that for all models, the adversarial attack mostly adds new edges. This indicates that edge insertion is stronger than edge deletion. Strong adaptive attacks on GNNGuard and Soft-Median-GDC seem to require the most edge deletions. Moreover, deletions are of much greater importance for local attacks.

## M Spectral properties of adaptive attacks

Previous studies have shown that adversarial attacks tend to focus the high-frequency (i.e., less significant) singular values of the adjacency matrix, both in the local [12] and global [30] setting. In consequence, defenses that exploit this observation to subdue attacks have been proposed (including SVD-GCN and ProGNN). This is a prime example of where (1) defenses were designed to circumvent specific attack characteristics and (2) an intuitive explanation exists of why the defense should improve robustness. However, our adaptive attacks have shown that neither (1) nor (2) entail actual robustness. In the case of SVD-GCN, it seems like the model becomes even less robust. It is only natural to ask whether our attacks exhibit spectral properties different from the high-frequency observation upon which SVD-GCN is built.

In Fig. M.1, we show the spectra of adjacency matrices before and after attacking GCN and SVD-GCN in various settings. Indeed, our adaptive attacks on SVD-GCN perturb more of the low frequencies and less of the high frequencies compared to attacks on GCN. Even though such low frequency-heavy perturbations are hypothesized to be "noticeable" [12, 30], it is unclear how this can be exploited in practice without knowing the clean graph or the underlying distribution of the spectrum. In § A, we give additional reasons why we disregard constraints beyond the $L_0$ difference.

Fig. M.1 also shows that, in contrast to previous beliefs, effective attacks on a GCN may lie in the low-frequency spectrum (see subplots a and c). This questions the strategy of dampening high-frequency singular values to defend against attacks in the first place.

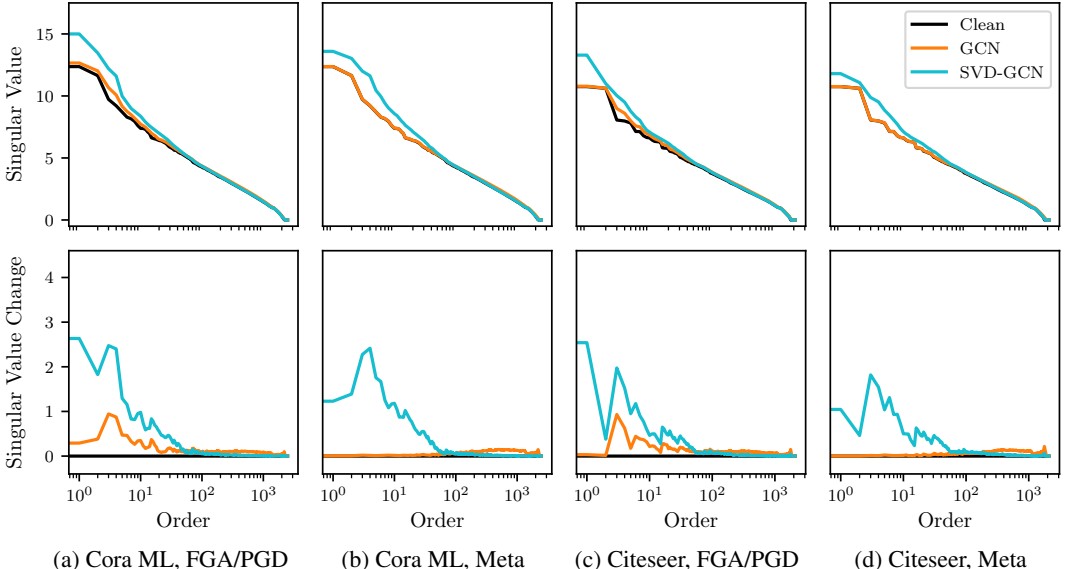

(a) Cora ML, FGA/PGD     (b) Cora ML, Meta     (c) Citeseer, FGA/PGD     (d) Citeseer, Meta

Figure M.1: Singular value spectra of the adjacency matrix before and after perturbation via global adaptive attacks with relative budget of 7.5% against GCN and SVD-GCN. Results are split into native evasion attacks (via FGA and PGD) and native poisoning attacks (via Metattack and Meta-PGD), and averaged in each group. The top row shows the absolute spectrum, and the bottom row the difference to the clean spectrum. The order is plotted logarithmically. We observe that attacks against SVD-GCN strongly perturb the low-order singular values, and it is evident from the relative plots that high-order singular values are perturbed less compared to attacks against GCN.

# N   On the scalability of adaptive attacks

In our main paper, we do not study adversarial robustness on larger graphs as (a) most defenses do not scale well and (b) we do not want to distract from our finding that structure defense evaluations are overly optimistic. Nevertheless, we consider scalability to be an important aspect for robustness as it is relevant for many applications. As mentioned in § 7, Geisler et al. [17] already study adaptive attacks scaled to large graphs. However, their work is focused on their own defense, and they only consider evasion. For these reasons, we now briefly discuss adaptive attacks on larger graphs.

In Fig. N.1, we show an adaptive attack against "Cosine-GCN" on arXiv from the Open Graph Benchmark [23] (169k nodes). Our Cosine-GCN defense is a natural equivalent of Jaccard-GCN [48] for continuous features. Similarly to Jaccard-GCN on the smaller graphs, Cosine-GCN also comes with some robustness w.r.t. a non-adaptive attack. However, once we apply an adaptive attack, it performs actually slightly worse than the GCN baseline.

**Scaling first order attacks.** The biggest challenge is certainly that the number of elements in the adjacency matrix scales quadratically with the number of nodes. One way to circumvent this "curse of dimensionality" is to use randomization. For our adaptive attack, we adopt Projected Randomized Block Coordinate Descent (PRBCD) [17]. PRBCD uses the same relaxation as PGD (see § 2 and § A). In each iteration of the attack, it considers only a random subset of edges for gradient update and subsequent projection. Then, for the next iteration, PRBCD keeps edges of high weight and randomly re-samples the edges of low weight. This way, the overhead remains constant in the block size. Since PRBCD is a first-order attack, it is natively adaptive for differentiable models.

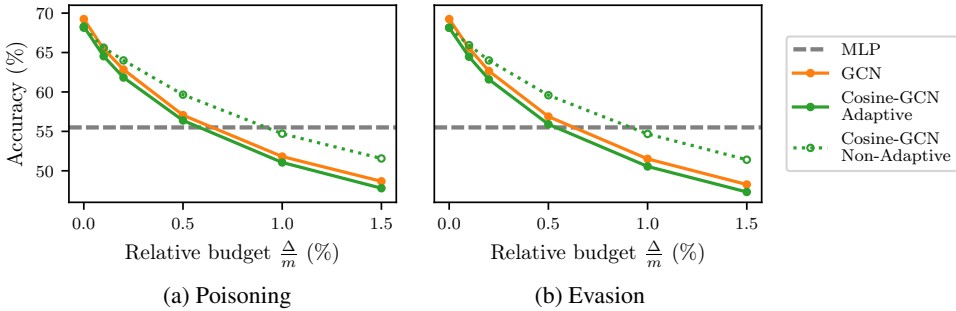

(a) Poisoning        (b) Evasion

Figure N.1: Adversarial accuracy on the large arXiv dataset per budget for the scalable PRBCD attack against a regular GCN and our Cosine-GCN (single random seed). We use a block size of 1 million edges and run the attack for 200 epochs. Thereafter, we keep the best block for another 50 epochs fixed. Poisoning is conducted by transferring perturbations from evasion.

**Evasion vs. poisoning.** Gradient-based poisoning attacks seem inherently more challenging since we need to unroll the training. Nevertheless, as long as we can run an evasion attack, there is the possibility to transfer the perturbed adjacency matrix to the poisoning setting. Here, we chose this approach. Still, Zügner & Günnemann [66] show in their appendix that only very few training steps are actually required for Metattack to be effective. Using a low number of training steps is therefore something to consider to scale direct poisoning attacks on larger graphs.