# OpenReview forum: "Are Defenses for Graph Neural Networks Robust?"
_NeurIPS.cc/2022/Conference — NeurIPS 2022 Accept_

### Official Review · Reviewer_pAxc · 2022-07-11

**Rating:** 7
**Confidence:** 4
**Soundness:** 3 good
**Presentation:** 3 good
**Contribution:** 3 good

**Summary:**

This work has revealed a serious issue of adversarial defenses on GNN: existing defense GNN models can be easily broken by applying customized adaptive attacks. Specifically, authors first categorize $45$ defense methods in literature, and develop proper methods to adaptively attack $7$ representative defense models. Experimental results indicate that the proposed adaptive attacks lower robustness of prior defense models by $40%$ on average. Moreover, authors also share some important lessons learnt from experiments and provide useful guidelines for evaluating new defense methods, which includes a robustness unit test produced in the experiments.

**Questions:**

  - How do authors handle the RAUC score when node features are also attacked (i.e., MLP accuracy will drop)?
  - Some prior works (e.g., SVD-GCN and ProGNN) have shown graph attacks are essentially high-rank attacks. How do adaptive attacks affect the graph spectrum? For instance, does the adaptive attack for SVD-GCN become a low-rank attack, so that it significantly drops SVD-GCN's accuracy? It would be great to see additional insights of adaptive attacks from the spectral perspective.

**Limitations:**

Authors mention the potential negative societal impact in a very vague way. Since the topic of this paper is highly related to the security of real GNN applications, a more detailed (e.g., a concrete example/scenario) could better illustrate the negative societal impact.

**Strengths And Weaknesses:**

**Key Strength**

  - The paper is clearly written. All the technical steps are easy to follow.
  - This work addresses a very important problem. Authors also propose a new metric for measuring the quality of attacks/defenses.
  - Authors have conducted extensive experiments to show how adaptive attacks, which include global/local attacks under evasion/poisoning setting, can break several representative defense GNN models.
  - The lessons and guidelines for adaptive attacking are reasonable. The proposed robustness unit test set is useful for evaluating new defense models.


**Key Weakness**

  - Authors only use two small datasets in experiments. Considering the scale of real GNN applications (e.g., [1]) for attacking, evaluating attacks/defenses on large graphs (with millions of nodes) is very important, as it may reveal more realistic lessons and guidelines. Note that there are a few scalable defense models such as [2].


[1]: Ying et al., "Graph Convolutional Neural Networks for Web-Scale Recommender Systems", KDD'18. \
[2]: Geisler et al., “Robustness of Graph Neural Networks at Scale”, NeurIPS'21.

---

> ### Author Response · Authors · 2022-08-02
> **Author Response to Reviewer pAxc**
>
> We thank the reviewer for the constructive feedback, notably the perspective on the importance of scalability and the suggestion to study the spectral properties of our attacks.
>
> ### Concerning scalability
>
> We agree that scalability is an important aspect that should also be considered for the robustness of GNNs. While there are some scalable defenses, the majority of the examined defenses is not. One reason why we decided against larger graphs, is that we strove for a setup to which defenses are applicable.
>
> However, acknowledging the importance of scalability in this context, we added a new section N to the appendix where we provide a brief case study for the Jaccard–GCN on how to design adaptive attacks at scale using the randomized first-order attack proposed in [2] (your numbering). Specifically, we look at the arXiv dataset from the Open Graph Benchmark and replace the Jaccard with the L2 distance to account for the continuous features.
>
> ### Concerning RAUC for feature perturbations (Question 1)
>
> As we only study structure perturbations, we did not discuss this in the original manuscript. However, we added a new section D on feature perturbations to the appendix that also answers this question. In short, one should resort to a strong baseline that does not use node features, e.g. label propagation.
>
> ### Concerning the spectral properties of adaptive attacks (Question 2)
>
> We added a new section M to the appendix where we empirically investigate your conjecture. In short, adaptive attacks on SVG-GCN indeed seem to focus on the low-frequency spectrum (i.e. most significant eigenvalues). On the other hand, attacks on GCN often perturb both low and high frequencies.
>
>
> ### Concerning security
>
> We will soon start the process of "responsible disclosure" – which has become the de facto standard after decades of debate on how to disclose security vulnerabilities [38] (now [39]). We will contact the authors of all examined defenses as they are most likely aware of affected parties or productive applications using these defenses. We will not make our work public before we have written consent from each involved party or a reasonable amount of time has passed. This approach is consistent with the closest related work in the image domain [38] and we will add a respective statement to the paper.
>
> We believe that our work will mostly influence researchers and practitioners to assess their models. As discussed in section 8, it is unlikely that a real-world adversary has perfect knowledge about data, labels and architecture. Our white-box threat model hinders the direct application of our methods by an external party. Moreover, we are fully convinced that works like ours are essential for the community to get past the state of security by obscurity.

---

> > ### Comment · Reviewer_pAxc · 2022-08-08
> > **Follow-up on authors' response**
> >
> > Thank you for your responses. I think most of my concerns have been addressed and tend to accept this paper.

---

### Official Review · Reviewer_fdqJ · 2022-07-11

**Rating:** 7
**Confidence:** 4
**Soundness:** 4 excellent
**Presentation:** 4 excellent
**Contribution:** 4 excellent

**Summary:**

This paper performs a thorough robustness analysis of robust graph neural network models that counteract adversarial attacks. It provides custom adaptive attacks to assess the robustness of robust graph neural networks and outlines the lessons for successfully designing such attacks. Extensive experiments demonstrate that the robustness of existing defense methods is overestimated as they perform badly under the adaptive attacks. Furthermore,  the transferability experiments show that perturbed graphs generated from different GNNs can transfer to other GNNs which can provide a black-box unit test to assess a model’s robustness.



**Questions:**

Q1 Can you  provide some explanation about why the attacks generated from other GNNs do not transfer well to SVD-GCN?

Q2 Have you tried the ensemble of the attacks, as what is done in [38]. For example, try the ensemble of the attacks generated from GCN and SVD-GCN, which could form a powerful attack with strong transferability.


**Strengths And Weaknesses:**

The strengths of this work are as follows.
1) The research problem is important. The evaluation of robust graph neural networks against adaptive attacks is rarely explored. Such evaluation protocol can help the community reliably assess model robustness.
2) The observations made in this paper are interesting. For example, the robustness of existing GNN defenses is overestimated and they can be broken under adaptive attacks.
3) The transferability experiment shows that the proposed method can also offer a black-box attack to assess a model’s robustness.
4) The paper is relatively well-written and the ideas are clearly clarified.

There are also some concerns that need to be addressed:
1) Compared with the adaptive attack in vision[38], the proposed method seems to requiring the tuning of hyper-parameters, which could be computationally expensive.
2) The authors are encouraged to provide some explanation about why the attacks generated from other GNNs do not transfer well to SVD-GCN?
3) It would be of interest to try the ensemble of the attacks, as what is done in [38]. For example, try the ensemble of the attacks generated from GCN and SVD-GCN, which could form a powerful attack with strong transferability.

---

> ### Author Response · Authors · 2022-08-02
> **Author Response to Reviewer fdqJ**
>
> We thank for the reviewer's encouraging feedback and for bringing up aspects like ensemble attacks and the frequency spectrum characteristics of attacks on SVD-GCN.
>
> ### Concerning hyperparameter tuning
>
> We agree with the reviewer that adaptive attacks do not constitute a "free lunch". However, Tramer et al. [38] (now [39]) also do not provide a one size fits all solution that can be applied effortlessly to a new defense (which potentially was designed with the sole focus of bypassing some previous attacks). For deep feed-forward neural networks, it is also well-known that solving the adversarial attack problem is NP-complete. For this reason and following the informally introduced "no-free lunch theorem" in [38], it is questionable whether there exists an efficient approximate attack that is *adaptive* to all possible defenses – also in the GNN domain. In [38]'s discussion, the authors also compare to a recent black-box attack, which would not require hyperparameter tuning but turns out to be weaker than their adaptive attacks.
>
> More practically, we also want to highlight that we did not spend excessive amounts of time tuning the attack hyperparameters, as is evident from our effort discussion (cf. lines 212-216, now 213-217). Also note that the development of adaptive attacks and the tuning of their hyperparameters required only little computational resources in comparison to the large-scale experiments then conducted with fixed hyperparameters.
>
> ### Concerning the transferability of attacks to SVD-GCN (Question 1)
>
> The edge perturbations required for an effective attack are very different between SVD-GCN and the other defenses. This is, for example, evident in Fig. J.1 (now L.1) a and b.
>
> Most importantly, attacks against other models often crucially perturb the high-frequency (i.e. least significant) eigenvalues of the adjacency matrix, which correspond to high frequencies in the spectrum. This property is exploited by SVD-GCN: it simply filters out the high-frequency spectrum (i.e. uses a low-pass filter). Hence, against SVD-GCN one needs specially crafted attacks that do not spend their budget on edges that are being filtered out.
>
> Following your feedback, we added a new section M to the appendix where we empirically study the spectra of the perturbed adjacency matrices generated by adaptive attacks on GCN and SVD-GCN.
>
> ### Concerning ensembles of attacks (Question 2)
>
> In fact, for most reported results, we already transfer the adversarial examples between models by default. We explain the details in lines 251-253 (now 252-255). As we mention there as well, these cross-model transfer attacks do not impact our evaluation much as adaptive attacks are almost always stronger (see also the main diagonal in Fig. 7).
>
> We agree that our accumulated collection of perturbed adjacency matrices could form a powerful attack. On these grounds we propose a robustness unit test in section 6: to indicate robustness, one should be robust at least to *all* the perturbed adjacency matrices of our study. Inspired by your suggestion, we also added a new section G to the appendix where we investigate which *subset* ensemble of models supplies the strongest transfer attack against our tested defenses.

---

> > ### Comment · Reviewer_fdqJ · 2022-08-07
> > **Thanks for the response**
> >
> > Thank you for the efforts in answering my questions. I tend to accept this paper and will keep my score as it is.

---

### Official Review · Reviewer_KkYT · 2022-07-11

**Rating:** 6
**Confidence:** 4
**Soundness:** 3 good
**Presentation:** 3 good
**Contribution:** 3 good

**Summary:**

The paper suggests that non-adaptive attacks lead to an overstate on adversarial robustness, and thus the authors recommend using adaptive attacks as a gold-standard. On GNN classifiers, the adversarial structure perturbation is to perturb the adjacency matrix from an evasion attack (after training with fixed model weights) or poisoning attack (before training). Also, adaptive and transferring attacks are compared and the former is stronger. With independency of budget and a baseline of an MLP, the experiments show that their adaptive attacks lower robustness by 40% in average across 7 defenses. The authors provide guidelines to design strong adaptive attacks.

**Questions:**

See weakness above.

**Strengths And Weaknesses:**

**Strengths:**

1. The paper provided a thorough review and analysis on existing GNN defense methods.

2. The paper provides guidelines to design strong adaptive attacks.

3. There are clear comparisons between non-adaptive and adaptive attacks and between poisoning and evasion attacks. The comparison strongly suggests that adaptive attacks should be a new standard in this area.

4. The experimental design is careful and comprehensive.

**Weakness:**

1. The topic of this paper seems to be overly general. As stated in lines 61-62, the attacks mentioned in the paper only focus on structure perturbations (i.e. by flipping an edge in adjacency matrix $A$). Feature perturbations (i.e. attacks on $X$) are not studied in this paper. But considering the strong conclusion claimed in the paper, these conditions may also need to be evaluated.  In “finding 4” (line 307), the author stated that they cannot confirm a trade-off between accuracy and robustness. As the weakness mentioned above, this trade-off may become perceptible if feature perturbations are introduced.



2. The attack budget is set to 15% of the total number of edges in the dataset which is a little bit tough for existing defense methods.



3. The designed adaptive attack is still highly related to existing works (PGD[47] and Metattack[60]).

---

> ### Author Response · Authors · 2022-08-02
> **Author Response to Reviewer KkYT**
>
> We thank for the reviewer's thorough feedback and, in particular, the critical thoughts on feature perturbations and ambiguities regarding attack budgets.
>
> ### Concerning feature perturbations
>
> We agree that feature perturbations are also of high importance. The main reason why we decided to study only structure perturbations is that defenses for GNNs rarely consider feature perturbations. For example, 6 out of the 7 examined defenses – which have been selected based on general popularity – do not target feature perturbations.
>
> The only examined defense that does study feature robustness is SVD-GCN [12]. Notably, due to [12]'s joint evaluation of feature and structural robustness, it is not clear whether SVD-GCN is sufficiently capable of defending against feature attacks (note that Nettack is biased towards structure perturbations). In our preliminary experiments, SVD-GCN did not indicate meaningful robustness gains compared to an undefended GCN even when considering non-adaptive feature perturbations.
>
> In the paper, we now emphasize stronger that we exclusively consider structure perturbations. Nevertheless, to highlight the importance of feature perturbations in the context of GNNs and to affirm that future feature-based defenses should likewise be evaluated using adaptive attacks, we also added an extra section D to the appendix and refer to it early on. In this new section, we discuss SVD-GCN [12] and important details on a proper evaluation of defenses for feature perturbations. For example, instead of the MLP baseline, one would need to compare to label propagation.
>
> We thank the reviewer for pointing out that "heading 4" could be misinterpreted. We refined the headings for the findings in section 5 to clarify that they refer to structure perturbations.
>
> We hope that these changes fully cover the reviewer's comments and are more than happy to address further suggestions.
>
> ### Concerning large attack budgets
>
> Please note that we determine the actual admissible budget range relatively to an MLP (i.e. consider budgets where the GNN outperforms an MLP). Specifically, for a GCN on Citeseer, we study budget ranges of up to 2\% and 4\% for poisoning and evasion respectively (6\% and 13\% on Cora ML).
>
> These ranges come from the definition of our RAUC metric (cf. lines 265-275, now 267-277). They are also apparent from the intersection points with the dashed gray MLP line in Fig. 4 and Fig. E.2 (now F.2). From these figures, it furthermore becomes clear that the range of sensible budgets depends on the dataset, model, and attack scenario (i.e. evasion vs. poisoning).
>
> Our reasoning (cf. lines 278-280, now 280-282) is that once perturbations are expected to be so strong that the accuracy drops below an MLP's accuracy, one should also resort to an MLP. The 15\% then only constitute a technical upper limit which is never reached in our evaluation (except for Soft-Median-GDC on Cora ML evasion).
>
> We amended section 5 to improve clarity about that.
>
> ### Concerning the relation to PGD and Metattack
>
> As we demonstrate, the efficacy of existing attacks is surprisingly high, given how poorly previous works utilize them to show the merit of their defenses w.r.t. structure perturbations. As such, our goal in this paper is not to design a brand new attack, but rather to show that one should use *adaptive* versions of existing simple attacks for proper robustness evaluation.
>
> We therefore argue that the relation to PGD [47] and Metattack [60] is not a weakness, but a strength: we show that by just making these existing attacks adaptive, both we and future researchers who strive to evaluate their defenses are able to obtain strong attacks. There is simply no need to develop entirely novel and specialized techniques. We believe that our findings, our methodology, and the resulting more realistic evaluation of robustness is highly beneficial to future research.

---

> > ### Comment · Reviewer_KkYT · 2022-08-08
> > **Thanks for the response**
> >
> > I appreciate the authors' careful response and my concerns are well-addressed. I will raise my score to weak accept.

---

### Author Response · Authors · 2022-08-02
**Meta Response**

To address the helpful feedback from all reviewers, we added the following sections to the manuscript:
- section D: feature perturbations (reviewers KkYT and pAxc)
- section G: additional aspects of transferability between ensembles of models (reviewer fdqJ)
- section M: spectral properties of adaptive attacks on GCN vs. SVD-GCN (reviewer fdqJ and pAxc)
- section N: discussion of scalability and an experiment of the arXiv graph with 169k nodes (reviewer pAxc)

---

### Meta-Review · Area_Chair_AUeC · 2022-08-23

**Recommendation:** Accept
**Confidence:** Certain

**Metareview:**

The recommendation is based on the reviewers' comments, the area chair's personal evaluation, and the post-rebuttal discussion.

This paper provides novel insights into the effectiveness of various defenses proposed for graph neural networks. All reviewers find the results convincing and valuable. The authors' rebuttal has successfully addressed the reviewers' concerns. Given the unilateral agreement, I am recommending acceptance


**Award:**

No

---

### Decision · Program_Chairs · 2022-09-14

Accept